# Accelerating Eigenvalue Dataset Generation via Chebyshev Subspace Filter

## Abstract

Eigenvalue problems are among the most important topics in many scientific disciplines. With the recent surge and development of machine learning, neural eigenvalue methods have attracted significant attention as a forward pass of inference requires only a tiny fraction of the computation time compared to traditional solvers. However, a key limitation is the requirement for large amounts of labeled data in training, including operators and their eigenvalues. To tackle this limitation, we propose a novel method, named **S**orting **C**hebyshev **S**ubspace **F**ilter (**SCSF**), which significantly accelerates eigenvalue data generation by leveraging similarities between operators—a factor overlooked by existing methods. Specifically, SCSF employs truncated fast Fourier transform (FFT) sorting to group operators with similar eigenvalue distributions and constructs a Chebyshev subspace filter that leverages eigenpairs from previously solved problems to assist in solving subsequent ones, reducing redundant computations. To the best of our knowledge, SCSF is the first method to accelerate eigenvalue data generation. Experimental results show that SCSF achieves up to a $3.5\times$ speedup compared to various numerical solvers.

## 1 Introduction

Solving eigenvalue problems is an important challenge in fields such as quantum physics [37], fluid dynamics [44], and structural mechanics [53]. Traditional numerical solvers, such as the Krylov-Schur algorithm [49], often suffer from prohibitively high computational costs when tackling complex problems. To overcome these computational challenges, recent advancements in deep learning [45, 28, 32] have demonstrated remarkable success as one forward pass only necessitates a tiny fraction of the computation time compared to numerical solvers, often in milliseconds.

Despite their success, data-driven approaches face a fundamental limitation: the reliance on labeled datasets. Training neural networks requires large-scale labeled data, which is often generated using computationally expensive traditional methods. It usually takes dozens of hours or even days. For example, the QM9 dataset [40] contains $1.34 \times 10^5$ molecular data points, each produced by solving Hamiltonian operator eigenvalue problems. These calculations typically employ traditional algorithms, whose computational costs can escalate dramatically with increasing problem complexity, like finer grid resolutions or higher accuracy requirements. This scalability issue represents a significant bottleneck for generating the labeled data needed to train deep learning models. Furthermore, the diversity of scientific problems leads to the need for a unique dataset for each scenario, which further intensifies this challenge of computational intractability. As a result, the high computational expense of generating eigenvalue data severely limits the application of data-driven approaches [57].

In particular, the dataset generation process typically involves six key steps, as illustrated in Figure 1 (left). Among these steps, solving the eigenvalue problem is the most computationally demanding

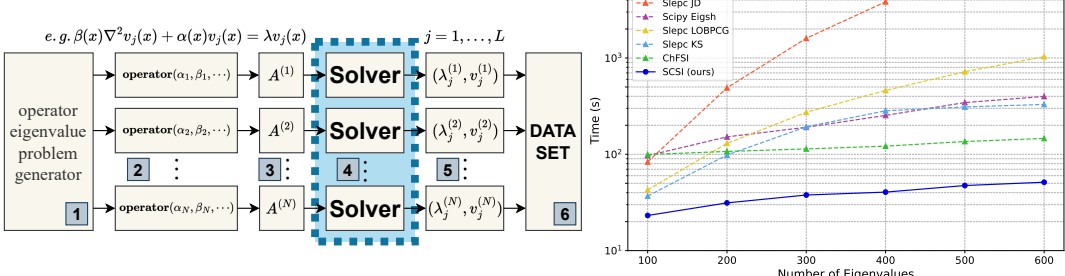

Figure 1: **Left.** Generation process of the eigenvalue dataset: 1. Generate a set of random problem parameters. 2. Derive the corresponding operators based on these parameters. 3. Convert the operators into matrices using discretization methods. 4. Independently solve for the matrix eigenvalues using numerical solvers. 5. Obtain the matrix eigenpairs, converting them into the operator eigenpairs. 6. Assemble the dataset. **Right.** Results of average computation times across various algorithms based on the number of eigenvalues solved on the Helmholtz operator dataset.

(step 4), accounting for 95% of the total processing cost [18]. Existing data generation methods typically compute the eigenvalues of each matrix in the dataset independently. However, operators in the dataset often share similarities, as they describe related physical phenomena, which can largely simplify and accelerate the eigenvalue-solving process. Existing approaches, however, fail to leverage these similarities, leading to significant computational redundancy. Previous works [52, 11] have demonstrated the potential of leveraging similarity to significantly reduce generation time of linear system datasets. However, how to effectively exploit matrix similarity to accelerate eigenvalue datasets generating remains an unknown problem.

To address this problem, we introduce a novel data generation approach, named **S**orting **C**hebyshev **S**ubspace **F**ilter (**SCSF**). SCSF is designed to use the eigenpairs of similar problems to reduce redundant computations in the eigenvalue solving process, thereby accelerating eigenvalue dataset generation. Specifically, at the beginning, SCSF employs a sorting algorithm based on truncated Fast Fourier transform (FFT), which arranges these problems efficiently, enhancing the adjacent correlation between problems in the queue and laying the groundwork for sequential solving. Then, SCSF accelerates the convergence of iterations and significantly reduces computation times by constructing a Chebyshev subspace filter, which solves the problem aided by the eigenpairs from previous problem solving. The core design of SCSF is to identify and exploit the close spectral distributions and invariant subspaces within these eigenvalue problems. SCSF coordinates the sequential resolution of these systems rather than treating them as discrete entities. This improved approach not only alleviates the computational demands of the eigenvalue algorithm but also significantly speeds up the generation of training data for data-driven algorithms. We summarize our contributions as follows:

- To the best of our knowledge, SCSF is the first method to accelerate the operator eigenvalue data generation.

- By using truncated FFT sorting and the Chebyshev filtered subspace iteration, we introduce a novel approach that transforms dataset generation into sequence eigenvalue problems.

- Comprehensive experiments demonstrate that SCSF substantially reduces the computational cost of eigenvalue dataset generation. As demonstrated in Figure 1 (right), our method achieves up to a $3.5\times$ speedup compared to state-of-the-art solvers.

## 2 Preliminaries

### 2.1 Discretization of Eigenvalue Problem

Our main focus is on solving the matrix eigenvalue problem, the most time-consuming part of eigenvalue data generation. As shown in Figure 1 (left), these problems are typically solved by numerical discretization methods such as FDM [50, 27]. These discretization techniques embed the infinite-dimensional Hilbert space of operators into an appropriate finite-dimensional space, thereby transforming operator eigenvalue problems into matrix eigenvalue problems. We provide a simple

example to clarify the discussed processes. A detailed process can be found in Appendix B. Specifically, we discuss the case that uses FDM to solve the eigenvalue problem of the two-dimensional Poisson operator, transforming it into a matrix eigenvalue problem:

$$k(x,y)\nabla^2 u(x,y) = \lambda u(x,y). \tag{1}$$

We map the problem onto a $2 \times 2$ grid (i.e., $N_x = N_y = 2$ and $\Delta x = \Delta y$), where both the variable $u_{i,j}$ and the coefficients $k_{i,j}$ follow a row-major order. This setup facilitates the derivation of the matrix eigenvalue equation:

$$\begin{bmatrix} k_{1,1} & 0 & 0 & 0 \\ 0 & k_{1,2} & 0 & 0 \\ 0 & 0 & k_{2,1} & 0 \\ 0 & 0 & 0 & k_{2,2} \end{bmatrix} \begin{bmatrix} -4 & 1 & 1 & 0 \\ 1 & -4 & 0 & 1 \\ 1 & 0 & -4 & 1 \\ 0 & 1 & 1 & -4 \end{bmatrix} \begin{bmatrix} u_{1,1} \\ u_{1,2} \\ u_{2,1} \\ u_{2,2} \end{bmatrix} = \lambda \begin{bmatrix} u_{1,1} \\ u_{1,2} \\ u_{2,1} \\ u_{2,2} \end{bmatrix}. \tag{2}$$

By employing various methods to generate the parameter matrices $P = \begin{bmatrix} k_{11} & k_{12} \\ k_{21} & k_{22} \end{bmatrix}$. Such as utilizing Gaussian random fields (GRF) or truncated polynomials, we can derive Poisson operators characterized by distinct parameters.

Typically, training a neural network requires a number of data from $10^3$ to $10^5$ [30]. Such a multitude of eigenvalue systems, derived from the same distribution of operators, naturally exhibit a highly similarity [48]. It is precisely this similarity that is key to the effective acceleration of SCSF. We can conceptualize this as the task of solving a sequential series of matrix eigenvalue problems:

$$A^{(i)}v_j^{(i)} = \lambda_j^{(i)}v_j^{(i)}, \quad j = 1, \cdots, L; \quad i = 1, 2, \cdots, N \tag{3}$$

where $L$ is the number of eigenvalues to be solved, $N$ is the number of eigenvalue problems, the matrix $A^{(i)} \in \mathbb{C}^{n \times n}$, the eigenvector $v_j^{(i)} \in \mathbb{C}^n$, and the eigenvalue $\lambda_j^{(i)} \in \mathbb{C}$ vary depending on the operator. We define the eigenpairs as $(\Lambda^{(i)}, V^{(i)})$, with $\Lambda^{(i)} = \text{diag}(\lambda_1^{(i)}, \ldots, \lambda_L^{(i)})$, $V^{(i)} = [v_1^{(i)}|\cdots|v_L^{(i)}]$, and $|\lambda_1^{(i)}| \leqslant |\lambda_2^{(i)}| \cdots \leqslant |\lambda_L^{(i)}|$.

## 2.2   The Chebyshev Polynomials and Chebyshev Filter

Chebyshev filtered subspace iteration is closely related to Chebyshev orthogonal polynomials [34, 41]. Chebyshev polynomials are widely used due to their strong approximation capabilities. The Chebyshev polynomials $C_m(t)$ of degree $m$ are defined on the interval $[-1, 1]$ and are expressed as

$$C_m(t) = \cos(m \cos^{-1}(t)), \quad |t| \leq 1. \tag{4}$$

$C_m(t)$ commonly referred to as the Chebyshev polynomial of the first kind, satisfies the following recurrence relation:

$$C_{m+1}(t) = 2tC_m(t) - C_{m-1}(t). \tag{5}$$

For a Hermitian matrix $A \in \mathbb{C}^{n \times n}$ and vectors $Y_0 \in \mathbb{C}^{n \times k}$, we use the three-term recurrence relation that defines Chebyshev polynomials in vector form:

$$C_{m+1}(Y_0) = 2AC_m(Y_0) - C_{m-1}(Y_0), \quad C_m(Y_0) \equiv C_m(A)Y_0. \tag{6}$$

The computation of $C_m(Y_0)$ and the Chebyshev filter is described in Algorithm 1. Let $A'$ denote the previously solved related matrix, with $(\lambda_i', v_i')$ in ascending order, and $\{\lambda_2', \ldots, \lambda_L'\} \in [\alpha, \beta]$. In Algorithm 1, the parameter $\lambda$ is typically approximated by $\lambda_1'$, while $c = \frac{\alpha+\beta}{2}$ and $e = \frac{\beta-\alpha}{2}$ represent the center and half-width of the interval $[\alpha, \beta]$, providing estimates for the spectral distribution of $A$.

---

**Algorithm 1:** Chebyshev Filter [5]

---

**Input:** Matrix $A \in \mathbb{C}^{n \times n}$, vectors $Y_0 \in \mathbb{C}^{n \times k}$, degree $m \in \mathbb{N}$, and parameters $\lambda, c, e \in \mathbb{R}$.
**Output:** Filtered vectors $Y_m = C_m(Y_0)$, where each vector $Y_{m,j}$ is filtered with a Chebyshev polynomial of degree $m$.

1   $A = (A - cI_n)/e, \quad \sigma_1 = e/(\lambda - c)$;
2   $Y_1 = \sigma_1 AY_0$;
3   **for** $i = 1, \ldots, m-1$ **do**
4      $\sigma_{i+1} = 1/(2/\sigma_1 - \sigma_i)$;
5      $Y_{i+1,1:m-1} = Y_{i,1:m-1}, \quad Y_{i+1,m:k} = 2\sigma_{i+1}AY_{i,m:k} - \sigma_{i+1}\sigma_i Y_{i,m:k}$;

---

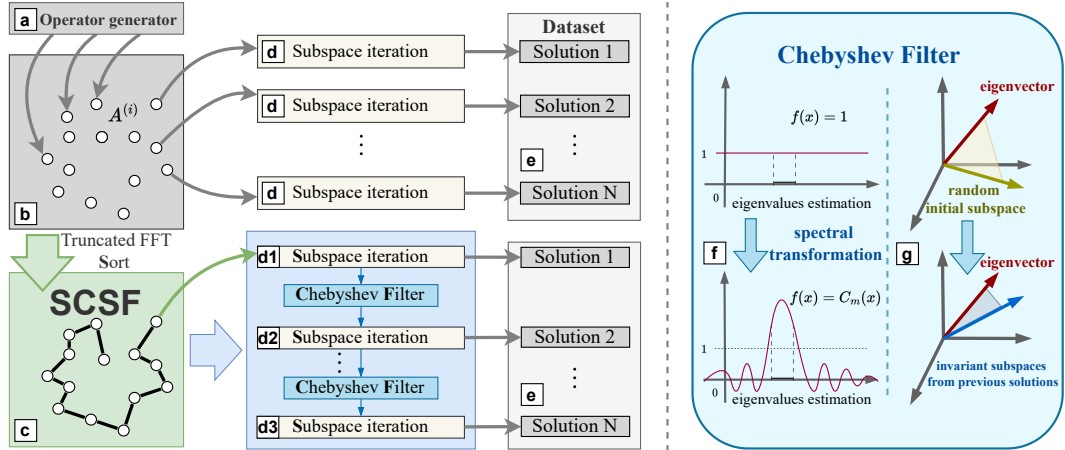

Figure 2: Algorithm Flow Diagram: **a**. Generation of operators to be solved. **b**. Discretization of operators into matrices. **c**. Apply SCSF algorithm to sort matrices, obtaining a sequence with strong correlations. **d**. Other algorithms independently solve eigenvalue problems. **d1, d2, d3**. SCSF algorithm utilizes Chebyshev subspace iterations to sequentially solve the eigenvalue problems. **e**. Assembly of eigenvalue pairs into a dataset. **f**. Amplification of the interval of interest through spectral transformation. **g**. Replacement of initial subspaces with previously solved invariant subspaces.

## 3 Method

In this section, we introduce our novel method, named the sorting Chebyshev subspace filter (SCSF), a fast data generation approach that efficiently solves eigenvalue problems by leveraging intrinsic spectral correlations among operators. SCSF incorporates two key components: (1) a truncated fast Fourier transform (FFT)-based approach for efficiently sorting operator eigenvalue problems and (2) the Chebyshev filtered subspace iteration (ChFSI) employed for sequential solving. By integrating these components, SCSF can use spectral information from the previous eigenvalue problem solving to aid the next eigenvalue problem solving, thus accelerating the eigenvalue data generation.

We first introduce the sorting algorithm that leverages the spectral similarities and provides the time complexity analysis in Section 3.1 . Then we give an introduction to the Chebyshev filtered subspace iteration in Section 3.2. Figure 2 shows the overview of our SCSF. Generally, the truncated FFT sorting algorithm ensures that successive matrices in the sequence exhibit close relations. Then ordered sequence enables ChFSI to effectively utilize prior information, thereby accelerating the solution process [5].

### 3.1   The Sorting Algorithm

To benefit the successive solving sequence of the eigenvalue problem, we need a sorting algorithm that pulls matrices with similar spectral properties, like invariant subspaces, close enough in the solving sequence, so that solving the current matrix in sequence can be easily boosted by the previous solving. Recalling Section 2.1, eigenvalue problem, the matrix $A^{(i)}$, is generated from the parameter matrix $P^{(i)}$ [31, 28]. A naive strategy is to use the Frobenius distance of the parameter matrices $P^{(i)}$ to perform a greedy sort [52]. And by repeatedly fetching without reservation from the remaining matrix in the dataset, we can reorganize the solving sequence so that the successive solving can benefit from the re-ordered sequence.

However, the main computational cost of such a naive sorting algorithm arises from repeatedly calculating the distances between different matrices $P$, which is directly related to the matrix dimension—that is, the resolution of operators. Existing works [17, 28] have shown that the key variables that affect operators stem from the low-frequency components of the parameter matrices $P$, while high-frequency components often represent noise or irrelevant data. Based on this insight, to reduce computational overhead during sorting, we first perform a truncated FFT on the parameter matrices to extract the low-frequency information before sorting. We then sort by comparing the Frobenius distances between these low-frequency components.

---

**Algorithm 2:** The Truncated FFT Sorting Algorithm

---

**Input:** Sequence of eigenvalue problems to be solved $A^{(i)} \in \mathbb{C}^{n \times n}$, corresponding parameter matrix $P^{(i)} \in \mathbb{C}^{p \times p}$, $i = 1, 2, \cdots, N$, $p_0$ is the truncation threshold for low frequencies, and $P_{low}^{(i)} \in \mathbb{C}^{p_0 \times p_0}$.

**Output:** Sequence for eigenvalue problems $seq_{mat}$.

1 Initialize the list with sequence $seq_0 = \{1, 2, \cdots, N\}$, $seq_{mat}$ is an empty list;

2 Set $i_0 = 1$ as the starting point. Remove 1 from $seq_0$ and append 1 to $seq_{mat}$;

3 **for** $i = 1, \cdots, N$ **do**

4      Let $P_{low}^{(i)} = \mathrm{Trunc}_{p_0}\left(\mathrm{FFT}(P^{(i)})\right)$. Perform truncated FFT on matrix $P^{(i)}$ to extract low-frequency information;

5 **for** $i = 1, \cdots, N-1$ *and* $dis = 1000$ **do**

6      **for** *each $j$ in $seq_0$* **do**

7          $dis_j$ = the Frobenius norm of the difference between $P_{low}^{(i_0)}$ and $P_{low}^{(j)}$;

8          **if** $dis_j < dis$ **then**

9              $dis = dis_j$ and $j_{min} = j$;

10      Remove $j_{min}$ from $seq_0$, append $j_{min}$ to $seq_{mat}$ and set $i_0 = j_{min}$;

11 Get the sequence for eigenvalue problems $seq_{mat}$;

---

As shown in Algorithm 2, suppose we have $N$ eigenvalue problems, the parameter matrices $P^{(i)} \in \mathbb{C}^{p \times p}$, and the low-frequency truncated matrices $P_{low}^{(i)} \in \mathbb{C}^{p_0 \times p_0}$. The computational complexity of directly using a greedy algorithm is $\mathcal{O}(N^2 p^2)$. Our sorting algorithm's complexity consists of two main parts: 1. FFT Computation: The complexity of FFT is $\mathcal{O}(p^2 \log p)$ per matrix. For $N$ matrices, this totals $\mathcal{O}(N p^2 \log p)$. 2. Greedy Sorting: The subsequent greedy sorting algorithm has a complexity of $\mathcal{O}(N^2 p_0^2)$. Overall, the total complexity is $\mathcal{O}(N^2 p_0^2 + N p^2 \log p)$. Since $p_0 \ll p$ and $p \ll N$, our sorting algorithm effectively reduces computational cost.

## 3.2    Chebyshev Filtered Subspace Iteration

After the sorting algorithm, we obtain a sequence of eigenvalue problems that exhibit strong correlations between consecutive problems. We employ the Chebyshev filtered subspace itera- tion [33, 43, 54, 5] that leverages the eigenpairs $(\Lambda^{(i-1)}, V^{(i-1)})$ of the previous problem $A^{(i-1)}$ to accelerate the iterative convergence of the subsequent problem $A^{(i)}$, thereby significantly enhancing computational performance. We focus on the most common scenario in eigenvalue problems where the operator is self-adjoint; in this case, the corresponding matrix $A$ is Hermitian.

Algorithm 3 outlines the process of ChFSI for solving the $i$-th eigenvalue problem $A^{(i)}$ ($1 < i \leqslant N$) where $L$ eigenvalues need to be solved. The initial approximate invariant subspace $V^{(i-1)}$ and spectral distribution $\Lambda^{(i-1)}$ are derived from the eigenvectors and eigenvalues of the previous problem $A^{(i-1)}$ in the sequence. The parameter $m$ denotes the polynomial degree in the filter function, e.g., $m = 20$. For the first eigenvalue problem $A^{(1)}$ in the sequence, the initial iterative subspace $\tilde{V}_0$ and initial spectrum $\tilde{\Lambda}_0$ are randomly generated.

In `line 3`, the Chebyshev filter is applied using the vector form of Chebyshev polynomials; details can be found in the preliminaries Section 2.2. After the Chebyshev filtering step, the vector block $\tilde{V}_0$ spanning the invariant subspace may become linearly dependent. To prevent this, orthonormalization is performed (`line 4`) using QR decomposition based on Householder reflectors. `Line 5` computes the Rayleigh quotient of matrix $A^{(i)}$ using the orthonormalized $\tilde{V}_0$, projecting the eigenvalue problem onto a subspace that approximates the desired eigenspace. In `line 6`, the reduced eigenvalue problem is diagonalized, and the computed eigenvectors are projected back to the original problem. At the end of the Rayleigh-Ritz step, relative residuals of the computed eigenvectors are calculated; converged eigenpairs are locked, and non-converged vectors are set to be filtered again (`line 7`).

Assuming $m$ is the degree of the polynomial, $n$ is the dimension of the matrix $A$, and $L$ is the number of eigenvalues to be solved, the computational complexity per iteration comprises: 1. Chebyshev filter: $\mathcal{O}(mn^2 L)$ 2. QR factorization: $\mathcal{O}(nL^2)$ 3. Rayleigh-Ritz procedure: $\mathcal{O}(n^2 L + nL^2 + L^3)$

---

**Algorithm 3:** Chebyshev Filtered Subspace Iteration

---

**Input:** Eigenvalue problem $A^{(i)}$, eigenpairs $(\Lambda^{(i-1)}, V^{(i-1)})$ of the previous eigenvalue
      problem $A^{(i-1)}$ where $\Lambda^{(i-1)} = \mathrm{diag}(\lambda_1^{(i-1)}, \ldots, \lambda_L^{(i-1)})$,
      $V^{(i-1)} = [v_1^{(i-1)} | \cdots | v_L^{(i-1)}]$, and filter degree $m$.

**Output:** Wanted eigenpairs $(\Lambda^{(i)}, V^{(i)})$.

**1** Initialize empty arrays/matrices $(\tilde{\Lambda}, \tilde{V})$, and set $\tilde{\Lambda}_0 = \Lambda^{(i-1)}$, $\tilde{V}_0 = V^{(i-1)}$;

**2 repeat**

**3**     Apply Chebyshev filter: $\tilde{V}_0 = C_m(\tilde{V}_0)$;

**4**     Perform QR orthonormalization on $QR = [\tilde{V}|\tilde{V}_0]$;

**5**     Compute Rayleigh quotient $G = Q_0^\top A^{(i)} Q$;

**6**     Solve the reduced problem $GW = W\tilde{\Lambda}_0$, and update $\tilde{V}_0 = \tilde{V}_0 W$;

**7**     Lock converged eigenpairs into $(\tilde{\Lambda}, \tilde{V})$;

**8 until** *the number of converged eigenpairs* $\geq L$;

**9** Return eigenpairs $(\Lambda^{(i)}, V^{(i)}) = (\tilde{\Lambda}, \tilde{V})$ ;

---

4. Residuals check: $\mathcal{O}(n^2 L)$ . Since $m \gg 1$ and $n \gg L$, the Chebyshev filtering step is the most computationally intensive.

The acceleration of the Chebyshev filtered subspace iteration heavily depends on selecting approximate invariant subspaces and eigenvalues that promote rapid convergence in subsequent iterations. Proper sorting amplifies their impact, reducing the number of iterations required. This underscores the critical importance of the sorting algorithm in our method.

## 4 Experiment

### 4.1 Experimental Settings

To comprehensively assess the performance of our approach SCSF against other algorithms, we conducted extensive experiments, each simulating the generation of an operator eigenvalue dataset. We primarily compared the average computation times across different numbers of eigenvalues solved and various matrix sizes. These tests encompassed four distinct datasets and five mainstream eigenvalue solving algorithms, with SCSF consistently delivering commendable results. The detailed data is provided in Appendix D.1, and the related work is discussed in Appendix A.

**Baseline**. Our focus solves the eigenvalue problem of matrices derived from self-adjoint differential operators, typically consisting of large Hermitian matrices. We benchmarked against the following mainstream algorithms implemented in libraries widely used: 1. Eigsh from SciPy (implicitly restarted Lanczos method) [51, 26], 2. Locally optimal block preconditioned conjugate gradient (LOBPCG) algorithm from SLEPc [24, 16], 3. Krylov-Schur (KS) algorithm from SLEPc [49], 4. Jacobi-Davidson (JD) algorithm from SLEPc [46], 5. Chebyshev filtered subspace iteration (ChFSI) [5, 54] with random initialization. For detailed information, please refer to Appendix C.1.

**Datasets**. To explore the adaptability of the algorithm across different matrix types, we investigate four distinct operator eigenvalue problems: 1. Generalized Poisson operator; 2. Second-order elliptic partial differential operator; 3. Helmholtz operator; 4. Fourth-order vibration equation. For a thorough description of the datasets and their generation, please refer to Appendix C.2.

All experiments focus on computing the smallest $L$ eigenvalues in absolute value and their corresponding eigenvectors. For the runtime environment and experimental parameters, refer to Appendix C.3 and C.4. The hyperparameter analysis experiments and the running time of each part of SCSF can be found in Appendix D.4 and D.3.

We note that all experiments use relative residual as the metric for solution precision, with its definition provided in Appendix C.5. SCSF is a numerical algebra algorithm that allows for adjustable solution precision as needed. It is purely an acceleration technique and does not alter the solution results at the specified precision. The solution precision for all experiments is set to at least 1e-8, which is significantly higher than the typical relative error range of neural networks (1e-1 to 1e-5), making it

| Dataset | $L$ | | Eigsh | LOBPCG | KS | JD | ChFSI | SCSF (ours) |
|---|---|---|---|---|---|---|---|---|
| Poisson | 200 | | 14.20 | 73.03 | 23.76 | 270.2 | 24.00 | **12.85** |
| 2500 | 300 | | 26.27 | 151.5 | 45.95 | 920.8 | 38.03 | **25.61** |
| 1e-12 | 400 | | 36.86 | 265.3 | 72.32 | 2691 | 57.41 | **33.91** |
| Ellipse | 200 | | 41.82 | 139.2 | 61.77 | 414.3 | 43.90 | **24.08** |
| 4900 | 300 | | 62.47 | 264.1 | 110.5 | 1446 | 60.69 | **29.88** |
| 1e-10 | 400 | | 87.19 | 459.7 | 188.7 | 3386 | 67.13 | **34.60** |
| Helmholtz | 200 | | 151.7 | 129.9 | 98.34 | 489.6 | 107.1 | **31.31** |
| 6400 | 400 | | 253.5 | 460.4 | 283.0 | 3829 | 121.5 | **40.52** |
| 1e-8 | 600 | | 398.8 | 1031 | 329.6 | - | 146.2 | **51.32** |
| Vibration | 200 | | 397.9 | 333.7 | 272.0 | 1230 | 300.8 | **85.70** |
| 10000 | 400 | | 635.6 | 1170 | 768.8 | - | 310.5 | **107.2** |
| 1e-8 | 600 | | 1037 | 2716 | 857.8 | - | 382.3 | **131.4** |

Table 1: Comparison of average computation times (in seconds) for eigenvalue problems using various algorithms. The first row lists different algorithms, the first column details the datasets, including matrix dimensions and solution precisions (relative residual), and the second column shows the number of eigenvalues $L$ computed for each matrix. The best algorithm is in **bold**. The symbol '-' denotes the result of a method that fails to converge under the given setting.

effectively a ground truth. Therefore, the datasets generated by different numerical algorithms will not affect the training performance of neural networks.

## 4.2 Main Experiment

Table 1 showcases selected experimental data. From this table, we can infer several conclusions: First, across all settings, our SCSF consistently has the lowest computation cost. The most significant improvements appeared in the Helmholtz dataset, where SCSF demonstrated speedups of $8\times$, $20\times$, $6\times$, $95\times$, and $3.5\times$ compared to Eigsh, LOBPCG, KS, JD, and ChFSI algorithms, respectively. These results confirm that SCSF effectively reduces inherent redundancies in sequential eigenvalue problems, substantially accelerating operator eigenvalue dataset generation.

Moreover, as the number of eigenvalues $L$ solved per matrix increases, the speed advantage of SCSF over other algorithms becomes more pronounced. For instance, on the second-order elliptic operator dataset, when solving for 200 eigenvalues, SCSF is 2.5 times faster than the Krylov-Schur method and 5.5 times faster at 400 eigenvalues. This efficiency stems from SCSF inheriting approximate invariant subspaces from previous solutions, effectively leveraging available information to expand the initial search space. Consequently, SCSF requires minimal additional iterations as $L$ increases, resulting in modest computation time growth.

Besides, the performance disparity across different datasets is significant. For example, on the generalized Poisson operator dataset, SCSF is only about 10% faster than Eigsh, yet it leads by 4-7 times on the Helmholtz dataset. This difference can be attributed to the numerical properties of different operators and the matrix assembly formats, which directly influence algorithmic performance.

We also conducted additional experiments to show that the impact of the matrix dimension is also significant. Results are shown in Figure 3, SCSF performs noticeably better as matrix dimensions increase. Below the matrix dimension of 3600, SCSF and Eigsh show comparable efficiency. However, beyond 5000, SCSF significantly outperforms Eigsh and other algorithms. For more details about matrix dimension influence, we refer to the results in Appendix D.2.

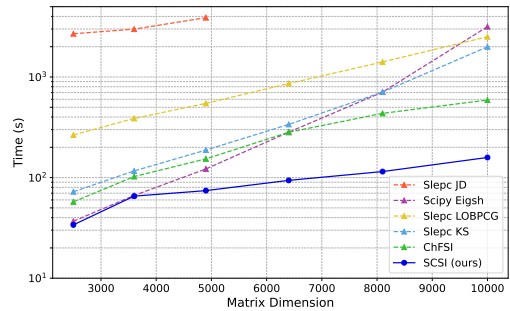

Figure 3: Plot of average computation time versus matrix dimension for solving 400 eigenvalues with a precision of 1e-12 on the generalized Poisson operator dataset.

This phenomenon can be analyzed from the matrix approximation of operators. For a fixed operator, its eigenvalues and eigenfunctions are fixed. Different matrix dimensions represent embedding the operator in different finite-dimensional linear spaces. For a fixed number of eigenvalues $L$, larger matrices more accurately approximate the true eigenvalues (the smallest $L$ eigenvalues by absolute value) of the operator. In other words, larger matrix dimensions result in fewer errors and noise in the computed eigenvalues, allowing for a clearer demonstration of the similarities between operators. Consequently, larger matrix dimensions allow SCSF to better exploit the similarities, yielding superior performance.

### 4.3 Efficacy of Chebyshev Subspace Filter

| $L$ | Eigsh | Eigsh* | LOBPCG | LOBPCG* | KS | KS* | JD | JD* | SCSF (ours) |
|---|---|---|---|---|---|---|---|---|---|
| 200 | 151.7 | 150.2 | 129.9 | 95.9 | 98.34 | 100.6 | 489.6 | 760.1 | **31.31** |
| 300 | 208.8 | 206.3 | 270.1 | 199.8 | 179.9 | 185.2 | 1803 | 3101 | **38.67** |
| 400 | 253.5 | 249.1 | 460.4 | 362.1 | 283.0 | 292.2 | 3829 | 6374 | **40.52** |
| 500 | 324.6 | 315.3 | 717.3 | 573.7 | 314.2 | 317.4 | - | - | **46.70** |
| 600 | 398.8 | 394.7 | 1031 | 866.0 | 329.6 | 335.7 | - | - | **51.32** |

Table 2: Impact of initial subspace modifications on average computation time (in seconds) for different algorithms. '*' denotes the modified version. The first row lists algorithms, and the first column shows the number of eigenvalues $L$ computed. The best algorithm is in **bold**, and '-' indicates the result of a method that fails to converge under the given setting.

To analyze the efficacy of the Chebyshev subspace filter, we conducted the following experiments. After sorting, the initial vector or subspace for the existing algorithms was set to the eigenvectors from the previous problem (the modified version '*'). We compared the computational time across different methods. All experiments were conducted on the Helmholtz operator dataset, with a matrix dimension of 6400 and a tolerance of 1e-8. The results are shown in Table 2.

First, the computation time for SCSF in all experiments was minimal, clearly demonstrating the efficacy of the Chebyshev subspace filter. This also highlights that the Chebyshev subspace filter is the optimal choice for leveraging problem similarity to reduce redundancy.

Second, modifying the initial setup had varying impacts on different algorithms. 1. LOBPCG: showed significant acceleration. Its underlying logic is similar to SCSF, both relying on iterative optimization of the subspace to solve the problem. The initial subspace has a considerable impact on the solution. 2. Eigsh and KS were almost unaffected. These methods start with an initial vector (rather than a subspace) and solve the problem through Krylov iteration. In other words, problem similarity only impacts a vector, with little effect on the overall time. 3. JD showed a performance decline. This is because its performance is sensitive to the size of the initial subspace. Our modification altered the default dimension of the initial subspace.

| $L$ | Time (s) | | Iteration | | Flops | | Filter Flops | |
|---|---|---|---|---|---|---|---|---|
| | w/o sort | sort | w/o sort | sort | w/o sort | sort | w/o sort | sort |
| 20 | 8.248 | 2.971 | 19.70 | 9.880 | 519.7 | 298.4 | 485.8 | 280.8 |
| 100 | 14.18 | 9.891 | 18.77 | 15.38 | 1984 | 1332 | 1798 | 970.1 |
| 200 | 18.45 | 12.85 | 36.30 | 33.67 | 4459 | 3944 | 3654 | 3192 |
| 300 | 34.59 | 25.61 | 47.50 | 39.18 | 8967 | 7544 | 6985 | 5702 |
| 400 | 42.60 | 33.91 | 47.43 | 45.18 | 12022 | 11182 | 9087 | 8338 |

Table 3: Performance comparison of SCSF with and without sorting. The first column lists the number of eigenvalues $L$ computed, while subsequent columns display average computation times, average iteration counts, total Flop counts, and filter Flop counts. Experiments used the matrix dimension of 2500 and precision 1e-12 on the generalized Poisson operator dataset.

## 4.4 Efficacy of Sorting Algorithms

We analyze the performance of the sorting algorithm module from two perspectives: 1. Comparing the performance of SCSF algorithm with and without 'sorting' as shown in Table 3. 2. Evaluating the effectiveness of different sorting algorithms as detailed in Tables 4 and 5.

We note that if the setting is 'w/o sort', SCSF is approximately equivalent to directly using the Chebyshev subspace filter. Unlike the ChFSI used in the main experiments, the initialization of each solve in the 'w/o sort' SCSF is set based on the information obtained from solving the previous problem (following the default unsorted sequence).

| Size | Greedy Total | Truncated FFT Sort (ours) | | |
|------|------|------|------|------|
| | | FFT | Greedy | Total |
| $10^2$ | 0.114 | 0.0016 | 0.0147 | 0.0163 |
| $10^3$ | 7.328 | 0.0164 | 1.421 | 1.438 |
| $10^4$ | 592.7 | 0.1658 | 150.9 | 151.1 |

Table 4: Comparison of average computation times (in seconds) for different sorting algorithms, with the first column indicating dataset size. Experiments used the matrix dimension of 6400 on the Helmholtz dataset.

Firstly, Table 3 indicates that incorporating sorting can improve SCSF speed to 1.3 to 2.8 times, reduce the number of iterations by 5% to 50%, and decrease total Flops by 7% to 43%. The effect of sorting is more pronounced with smaller numbers of solutions $L$. This is because when $L$ is large, the inherited subspace already contains most of the necessary correlation information, diminishing the impact of sorting. Moreover, the Flops in the Filter component constitute over 70% of SCSF's computational load. A detailed time analysis of different aspects of SCSF can be found in Appendix D.3. Additionally, the 'w/o sort' SCSF achieves a computational speedup of 1.2 to 1.5 times compared to the ChFSI used in the main experiments. The primary difference lies in their initialization strategies: ChFSI uses random initialization for each solve, whereas the 'w/o sort' SCSF leverages information from the previous problem for initialization. This indicates that, even without sorting, there is a certain level of similarity between problems in the dataset. Such similarity can effectively accelerate the solving process.

Secondly, as shown in Table 4, our designed truncated FFT sorting algorithm incurs significantly lower time cost compared to the complete greedy sorting in SKR [52], with its benefits becoming more pronounced as the dataset size increases. In the truncated FFT sorting algorithm, the FFT contributes minimally to computational overhead but significantly reduces the time required for subsequent greedy sorting. Table 5 shows SCSF solution times for matrices sorted using either greedy or truncated FFT sorting are nearly identical, highlighting its effectiveness.

| | w/o sort | Greedy | Ours |
|------|------|------|------|
| Time (s) | 66.66 | 40.52 | 40.52 |
| Iteration | 10.4 | 5.5 | 5.5 |

Table 5: Comparison of average computation times and iteration counts for different sorting algorithms using SCSF. Experiments used the matrix dimension of 6400 on the Helmholtz dataset, precision 1e-8, and targeting 400 eigenvalues.

Furthermore, our experiments show that as long as the truncated FFT is configured with reasonable parameters (e.g., truncating at $p_0 = 20$, where $p_0$ is much smaller than the dimension $p$ of the parameter matrix $P$), it achieves excellent performance without the need for a large $p_0$. For related experiments, please refer to Appendix D.4.3.

## 5 Limitations and Conclusions

In this paper, we introduced SCSF algorithm. To the best of our knowledge, this is the first method to accelerate eigenvalue dataset generation by reducing computational redundancy in the associated matrix eigenvalue problems. The proposed SCSF algorithm significantly reduces the computational overhead of eigenvalue dataset generation, thereby addressing a major obstacle to the application of neural networks in scientific computing.

However, several areas warrant further exploration: 1. While this paper primarily focuses on general eigenvalue problems, extending the method to nonlinear eigenvalue problems requires tailored designs to achieve optimal computational efficiency. 2. Within the sorting algorithm of SCSF, there is potential to identify more effective distance metrics tailored to specific operators, aiming to enhance the correlation and efficiency of the sorted eigenvalue problems.

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

# A Related work

## A.1 Eigenvalue Datasets and Neural Eigenvalue Methods

Eigenvalue datasets are widely utilized in neural eigenvalue methods. In molecular chemistry research, eigenvalue algorithms are commonly employed to determine critical molecular properties, such as orbital energy levels [23]. These properties form the foundation of datasets and are obtained by solving the eigenvalue problem of the Schrödinger equation and the Hamiltonian operator [15]. Prominent datasets in this domain include QM7 [7], QM9 [40], ANI-1 [47], and MD17 [10]. In materials science, eigenvalue algorithms are often applied to solve for electronic band structures and density of states in materials. Representative datasets in this field include the materials project [20] and OQMD [22]. These datasets have been extensively used to train and validate neural eigenvalue methods [45, 4, 42], driving advancements in molecular property prediction and materials design. In fluid dynamics and structural mechanics, eigenvalue algorithms are frequently utilized for modal analysis. Recently, many data-driven modal analysis algorithms have emerged, requiring eigenvalue datasets corresponding to differential operators for training [36, 19, 2, 9, 1]. Additionally, some studies leverage operator eigenvalue datasets to optimize algorithms. For instance, [32] accelerates the solution of linear systems by predicting the eigenfunctions of operators.

## A.2 Eigenvalue Data Generation Algorithms

Training data-driven algorithms require a large amount of labeled eigenvalue data. Typically, the generation of these high-precision data is obtained by traditional algorithms. In the field of computational mathematics, solving operator eigenvalue problems often involves utilizing various discretization methods such as finite difference methods (FDM) [50], finite element methods (FEM) [18, 21]. These discretization methods transform operator eigenvalue problems into matrix eigenvalue problems, which are then solved using the corresponding matrix algorithms. For larger matrices, the Krylov-Schur algorithm [49], Jacobi-Davidson [46], and locally optimal block preconditioned conjugate gradient (LOBPCG) [24] are among the most frequently employed algorithms [13].

Nonetheless, traditional methods were not designed for dataset generation, resulting in high computational costs, which have become a significant barrier to the advancement of data-driven approaches [57, 14]. Recent data augmentation research [8, 29] has led to the development of methods that preserve symmetries and conservation laws, enhancing model generalization and data efficiency. [52, 11] report acceleration in the process of solving linear equations, thereby speeding up the generation of PDE datasets.

However, these improvements largely focus on neural networks or the rapid solution of linear system-based PDEs, without discussing optimizations in the generation of eigenvalue datasets.

## A.3 Chebyshev Filter Technique

The Chebyshev filter technique originates from polynomial approximation theory, where the core concept involves using Chebyshev polynomials to accelerate the convergence of eigenvalues [59]. This technique constructs a polynomial filter that selectively amplifies spectral components in a specified interval, thereby speeding up the solution of specific eigenvalues. This technique is particularly effective in dealing with sequence eigenvalue problems [43, 60] and has been applied in various contexts, such as stability analysis in electronic structure [38, 3] and quantum chemical computations [35, 58, 61].

Due to the chaotic and disordered nature of eigenvalue problems in the dataset, directly applying the Chebyshev filter technique fails to accelerate dataset generation. To further adapt this technique to the generation of operator eigenvalue datasets, we have developed a specialized sorting algorithm that transforms dataset generation into sequence eigenvalue problems. Throughout the solving process, eigenpairs obtained from previous solutions are used to construct Chebyshev filters, accelerating subsequent solutions.

# B From Differential Operator to Matrix Eigenvalue Problem: An Example

## B.1 Overview

The general methodology for solving the eigenvalue problems of differential operators numerically, employing techniques such as Finite Difference Method (FDM), Finite Element Method (FEM), and Spectral Method, can be delineated through the following pivotal steps [50, 18, 21, 27]:

1. Mesh Generation: This step involves dividing the domain, over which the differential operator is defined, into a discrete grid. The grid could be composed of various shapes, including squares, triangles, or more complex forms, depending on the problem's geometry.

2. Operator Discretization: The differential operator is transformed into its discrete counterpart. Essentially, this maps the operator from an infinite-dimensional Hilbert space to a finite-dimensional representation.

3. Matrix Assembly: In this phase, the discretized operator is represented in a matrix form. For linear differential operators, this involves creating a system of matrix eigenvalue problems. For nonlinear operators, iterative methods akin to Newton's iteration are employed, transforming the problem into a sequence of matrix eigenvalue problems.

4. Applying Boundary Conditions: This involves discretizing and applying boundary conditions specific to the differential operator in question, which are then incorporated into the matrix system.

5. Solving the Matrix Eigenvalue Problem: This stage, often the most computationally intensive, entails solving the matrix for its eigenvalues and eigenvectors, which correspond to the eigenvalues and eigenfunctions of the original differential operator.

6. Obtaining the Numerical Solution: The final step involves mapping the obtained numerical solutions back onto the original domain, analyzing them for accuracy and stability, and interpreting them in the context of the initial problem.

## B.2 Example

To illustrate how the FDM can transform the wave equation into a system of matrix eigenvalue problems, let's consider a concrete and straightforward example. Assume we aim to solve a one-dimensional wave equation's operator eigenvalue problem, expressed as

$$-\frac{d^2u}{dx^2} = \lambda u,$$

over the interval $[0, L]$. The boundary conditions are $u(0) = u(L) = 0$, signifying fixed-end conditions. In this context, $u(x)$ denotes the eigenfunction, and $\lambda$ represents the eigenvalue.

1. Mesh Generation: Using the central difference quotient, we divide the interval $[0, L]$ into $N + 1$ evenly spaced points, including the endpoints. The distance between adjacent points is denoted as $\Delta x = \frac{L}{N}$.

2. Operator Discretization: This step involves formulating the difference equation. At each interior node, which excludes the endpoints and totals $N - 1$ points, we apply a central difference approximation for the second derivative, represented as

$$\frac{d^2u}{dx^2} \approx \frac{u(x_{i+1}) - 2u(x_i) + u(x_{i-1})}{(\Delta x)^2}$$

3. Matrix Assembly: In this phase, the discretized operator is represented in a matrix form. Following the approximation, we construct the matrix $A$, an $N - 1 \times N - 1$ tridiagonal matrix, crucial for the computations. The matrix $\boldsymbol{A}$ is constructed as:

$$\boldsymbol{A} = \frac{1}{(\Delta x)^2} \begin{bmatrix} -2 & 1 & 0 & \cdots & 0 \\ 1 & -2 & 1 & \cdots & 0 \\ 0 & 1 & -2 & \cdots & 0 \\ \vdots & \vdots & \vdots & \ddots & \vdots \\ 0 & 0 & 0 & \cdots & -2 \end{bmatrix}$$

4. Applying Boundary Conditions: For the wave equation with boundary conditions $u(0) = u(L) = 0$, these fixed-end conditions are integrated into the matrix equation. In the FDM framework, the values at the endpoints ($u_0$ and $u_N$) are zero, directly reflecting the boundary conditions. The impact of these conditions is encapsulated in the matrix $\boldsymbol{A}$, affecting the entries related to $u_1$ and $u_{N-1}$ (the grid points adjacent to the boundaries). The tridiagonal matrix $\boldsymbol{A}$ incorporates these boundary conditions, ensuring that the computed eigenfunctions satisfy $u(0) = u(L) = 0$.

5. Solving the Matrix Eigenvalue Problem: The final computational step involves solving the matrix eigenvalue problem, expressed as $\boldsymbol{Au} = \lambda \boldsymbol{u}$. This includes determining the eigenvalues $\lambda$ and corresponding eigenvectors $\boldsymbol{u}$, which are discrete approximations of the eigenfunctions of the original differential equation.

6. Obtaining the Numerical Solution: By solving the eigenvalue problem, we obtain numerical solutions that approximate the behavior of the original differential equation. These solutions reveal the eigenvalues and eigenvectors and provide insights into the physical phenomena modeled by the equation.

# C  Details of Experimental Setup

## C.1  Baseline

The baseline algorithms were implemented using the following numerical computing libraries:

- Eigsh: A SciPy (v1.14.1) implementation wrapping ARPACK's SSEUPD and DSEUPD functions, which compute eigenvalues and eigenvectors using the Implicitly Restarted Lanczos Method. The default parameters were used.

- Locally Optimal Block Preconditioned Conjugate Gradient (LOBPCG): Implemented in SLEPc (v3.21.1) with default parameters.

- Krylov-Schur (KS): Implemented in SLEPc (v3.21.1) with default parameters.

- Jacobi-Davidson (JD): Implemented in SLEPc (v3.21.1). The implementation uses 'bcgsl' as the linear equation solver, 'bjacobi' as the preconditioner, and sets the linear equation solving precision to 1e-5.

- ChFSI: Implemented in ChASE (v1.6) with default parameters.

## C.2  Dataset

1. Generalized Poisson Operator

We consider two-dimensional generalized Poisson operators, which can be described by the following equation [28, 39, 25, 31]:
$$-\nabla \cdot (K(x,y)\nabla h(x,y)) = \lambda h(x,y),$$
In our experiment, $K(x,y)$ is derived using the Gaussian Random Field (GRF) method. We convert these operators into matrices using the central difference scheme of FDM. The parameters inherent to the GRF serve as the foundation for our sort scheme.

2. Second-Order Elliptic Partial Differential Operator

We consider general two-dimensional second-order elliptic partial differential operators, which are frequently described by the following generic form [12, 6]:
$$\mathcal{L}u \equiv a_{11}u_{xx} + a_{12}u_{xy} + a_{22}u_{yy} + a_1 u_x + a_2 u_y + a_0 u = \lambda u,$$
where $a_0, a_1, a_2, a_{11}, a_{12}, a_{22}$ are constants, and $f$ represents the source term, depending on $x, y$. The variables $u, u_x, u_y$ are the dependent variables and their partial derivatives. The equation is classified as elliptic if $4a_{11}a_{22} > a_{12}^2$.

In our experiments, $a_{11}, a_{22}, a_1, a_2, a_0$ are uniformly sampled within the range $(-1, 1)$, while the coupling term $a_{12}$ is sampled within $(-0.01, 0.01)$. We then select equations that satisfy the elliptic condition to form our dataset. We convert these operators into matrices using the central difference scheme of FDM. The coefficients $a_0, a_1, a_2, a_{11}, a_{12}, a_{22}$ serve as the foundation for our sort scheme.

607 3. Helmholtz Operator

608 We consider two-dimensional Helmholtz operators, which can be described by the following equa-
609 tion [56]:

$$\nabla \cdot (p(x,y)\nabla u(x,y)) + k^2(x,y) = \lambda u(x,y),$$

610 Physical Contexts in which the Helmholtz operator appears: 1. Acoustics; 2. Electromagnetism; 3.
611 Quantum Mechanics.

612 In Helmholtz operators, $k$ is the wavenumber, related to the frequency of the wave and the properties
613 of the medium in which the wave is propagating. In our experiment, $p(x,y)$ and $k(x,y)$ are derived
614 using the GRF method. The parameters inherent to the GRF serve as the foundation for our sort
615 scheme.

616 4. Vibration Equation

617 We consider the vibration equation for thin plates, which can be described by the following eigenvalue
618 problem [55]:

$$\nabla^2\big(D(x,y)\nabla^2 u(x,y)\big) = \lambda\rho(x,y)u(x,y),$$

619 Physical contexts in which the vibration equation appears: 1. Structural dynamics of thin plates; 2.
620 Modal analysis in mechanical engineering; 3. Vibrational behavior of elastic materials.

621 In this equation, $D(x,y)$ represents the flexural rigidity of the plate, $\rho(x,y)$ is the density distribution,
622 and $\lambda$ corresponds to the eigenvalue, which is related to the natural frequencies of the system. The
623 eigenfunction $u(x,y)$ describes the mode shapes of vibration.

624 In our experiment, $D(x,y)$ and $\rho(x,y)$ are derived using the GRF method. The parameters inherent
625 to the GRF serve as the foundation for our sorting scheme.

626 ## C.3 Environment

627 To ensure consistency in our evaluations, all comparative experiments were conducted under uniform
628 computing environments. Specifically, the environments used are detailed as follows:

629 • Platform: Docker version 4.33.1 (windows 11)

630 • Operating System: Ubuntu 22.04.3 LTS

631 • Processor: CPU AMD Ryzen 9 8945HS w, clocked at 4.00 GHz

632 ## C.4 Experimental Parameter Configuration

633 All baseline methods were implemented using their default parameters from respective libraries.

634 For SCSF, the following configurations were adopted:

635 • The size of the inherited subspace varies according to the number of eigenvalues to be
636 computed. Specifically, when calculating 20, 100, 200, 300, and 400 eigenvalues, the
637 corresponding subspace sizes are set to 4, 20, 40, 60, and 80, respectively.

638 • The filter degree parameter $m$ is consistently set to 20 across all experiments.

639 • Truncation threshold for low frequencies $p_0$ is consistently set to 20 across all experiments.

640 • Each experiment generates a dataset consisting of 1,000 samples. In this paper, the Experi-
641 mental tables report the average solving time for each eigenvalue problem.

642 ## C.5 Error Metrics

643 • Relative Residual Error:
644 To estimate the bias of the eigenpair $(\tilde{v}, \tilde{\lambda})$ predictions, we employ relative residual error as
645 follows:

$$\text{Relative Residual Error} = \frac{||\mathcal{L}\tilde{v} - \tilde{\lambda}\tilde{v}||_2}{||\mathcal{L}\tilde{v}||_2}.$$

Here, $\tilde{v}$ represents the eigenfunction predicted by the model, and $\tilde{\lambda}$ denotes the eigenvalue predicted by the model. When $\tilde{\lambda}$ is the true eigenvalue and $\tilde{v}$ is the true eigenfunction, the Relative Residual Error equals $0$.

# D  Experimental Data and Supplementary Experiments

## D.1  Main Experimental Data

As shown in Tables 7, 6, 9, SCSF showed the best performance among all tested configurations

| $L$ | Eigsh | LOBPCG | KS | JD | ChFSI | SCSF (ours) |
|---|---|---|---|---|---|---|
| 150 | 9.15 | 46.8 | 14.9 | 138 | 17.3 | 7.95 |
| 200 | 14.2 | 73.0 | 23.8 | 270 | 24.0 | 12.9 |
| 250 | 19.8 | 109 | 34.3 | 553 | 30.2 | 19.0 |
| 300 | 26.3 | 152 | 45.6 | 921 | 38.0 | 25.7 |
| 350 | 31.5 | 203 | 58.4 | 1732 | 45.8 | 29.8 |
| 400 | 36.9 | 265 | 72.3 | 2691 | 57.4 | 33.9 |
| 450 | 42.8 | 342 | 87.3 | 3708 | 74.2 | 38.3 |

Table 6: Comparison of average computation times (in seconds) for eigenvalue problems using various algorithms on the generalized Poisson operator dataset. The first row lists different algorithms, and the first column shows the number of eigenvalues $L$ computed for each matrix. Matrix dimension = 2500, precision = 1e-12.

| $L$ | Eigsh | LOBPCG | KS | JD | ChFSI | SCSF (ours) |
|---|---|---|---|---|---|---|
| 150 | 31.35 | 91.80 | 40.65 | 214.80 | 38.37 | 19.62 |
| 200 | 41.82 | 139.20 | 61.77 | 414.30 | 43.90 | 24.08 |
| 250 | 52.17 | 197.04 | 84.65 | 861.44 | 53.42 | 28.00 |
| 300 | 62.47 | 264.10 | 110.50 | 1446.00 | 60.69 | 29.88 |
| 350 | 74.59 | 355.18 | 147.01 | 2324.88 | 64.94 | 31.52 |
| 400 | 87.19 | 459.70 | 188.70 | 3386.00 | 67.13 | 34.60 |
| 450 | 100.28 | 577.67 | 235.56 | 4629.38 | 76.32 | 40.05 |

Table 7: Comparison of average computation times (in seconds) for eigenvalue problems using various algorithms on the second-order elliptic operator dataset. The first row lists different algorithms, and the first column shows the number of eigenvalues $L$ computed for each matrix. Matrix dimension = 4900, precision = 1e-10.

| $L$ | Eigsh | LOBPCG | KS | JD | ChFSI | SCSF (ours) |
|---|---|---|---|---|---|---|
| 200 | 151.70 | 129.90 | 98.34 | 489.60 | 107.12 | 31.31 |
| 300 | 190.84 | 273.08 | 192.88 | 1601.08 | 113.73 | 37.78 |
| 400 | 253.50 | 460.40 | 283.00 | 3829.00 | 121.53 | 40.52 |
| 500 | 344.60 | 720.33 | 310.21 | - | 135.73 | 47.41 |
| 600 | 398.80 | 1031.00 | 329.60 | - | 146.24 | 51.32 |

Table 8: Comparison of average computation times (in seconds) for eigenvalue problems using various algorithms on the Helmholtz operator dataset. The first row lists different algorithms, and the first column shows the number of eigenvalues $L$ computed for each matrix. Matrix dimension = 6400, precision = 1e-8. The symbol '-' denotes data not recorded due to excessive computation times.

| $L$ | Eigsh | LOBPCG | KS | JD | ChFSI | SCSF (ours) |
|-----|-------|--------|------|------|-------|-------------|
| 200 | 397.9 | 333.7 | 272.0 | 1230 | 300.8 | 85.70 |
| 300 | 516.8 | 750.0 | 520.0 | 3600 | 305.0 | 96.50 |
| 400 | 635.6 | 1170 | 768.8 | - | 310.5 | 107.2 |
| 500 | 820.0 | 1950 | 810.0 | - | 350.0 | 120.0 |
| 600 | 1037 | 2716 | 857.8 | - | 382.3 | 131.4 |

Table 9: Comparison of average computation times (in seconds) for eigenvalue problems using various algorithms on the Vibration operator dataset. The first row lists different algorithms, and the first column shows the number of eigenvalues $L$ computed for each matrix. Matrix dimension = 10000, precision = 1e-8. The symbol '-' denotes data not recorded due to excessive computation times.

## D.2    Analysis of the Influence of Matrix Dimension

| Matrix Dimension | Eigsh | LOBPCG | KS | JD | ChFSI | SCSF (ours) |
|------------------|-------|--------|---------|---------|-------|-------------|
| 2500 | 36.86 | 265.30 | 72.32 | 2691.00 | 57.41 | 33.91 |
| 3600 | 66.41 | 387.20 | 116.50 | 2990.00 | 102.4 | 65.41 |
| 4225 | 89.13 | 467.74 | 151.36 | 3548.13 | 126.2 | 70.79 |
| 4900 | 121.90 | 546.20 | 187.80 | 3886.00 | 153.5 | 74.23 |
| 5625 | 186.21 | 691.83 | 251.19 | - | 216.8 | 85.11 |
| 6400 | 282.80 | 860.00 | 337.70 | - | 282.2 | 93.86 |
| 8100 | 707.95 | 1412.54 | 707.95 | - | 435.1 | 114.82 |
| 10000 | 3162.28 | 2511.89 | 1995.26 | - | 590.3 | 158.49 |

Table 10: Comparison of different algorithms' computation time (in seconds) for varying matrix dimensions using the generalized Poisson operator dataset. Results show average computation times for solving 400 eigenvalues with a precision of 1e-12.

As demonstrated in Table 10, the impact of matrix dimension on algorithm performance reveals several key insights. For matrices below dimension 3600, SCSF and Eigsh show comparable efficiency. However, SCSF's advantages become increasingly pronounced as matrix dimensions grow larger. At dimension 10000, SCSF achieves remarkable speedups: 20× faster than Eigsh, 16× faster than LOBPCG, 13× faster than KS, and 3.7× faster than ChFSI. This phenomenon can be attributed to how larger matrix dimensions result in fewer errors and noise in the computed eigenvalues, allowing SCSF to better exploit similarities between problems. Additionally, the JD algorithm becomes computationally intractable at and above dimension 5625, while SCSF maintains stable performance even at high dimensions.

## D.3    Analysis of Computational Times for SCSF Components

| All | Filter (line 3) | QR (line 4) | RR (line 5) | Resid (line 6) | Sort |
|-----|-----------------|-------------|-------------|----------------|------|
| 9.89e+0 | 7.41e+0 | 3.12e-1 | 9.76e-1 | 7.95e-1 | 1.51e-2 |

Table 11: Analysis of Computational Times (in seconds) for SCSF Components.

We conducted a statistical analysis of the average time consumption for each component of the SCSF algorithm on the generalized Poisson operator dataset, with a matrix dimension of 2500 and the number of eigenvalues to be solved set to 100. The results are presented in Table 11. The notation "line x" within parentheses corresponds to line x in Algorithm 3, "ALL" denotes the total time consumption, and "sort" represents the average time required by the sorting algorithm. It is evident

that the filtering process accounts for over 70% of the total time consumption, which aligns with our theoretical analysis in Section 3.2.

## D.4  Analysis of Hyperparameters

### D.4.1  Degree Parameter

| Deg | 12 | 16 | 20 | 24 | 28 | 32 | 36 | 40 |
|---|---|---|---|---|---|---|---|---|
| Time (s) | 43.92 | 39.79 | 40.52 | 40.64 | 40.85 | 41.13 | 41.19 | 43.50 |

Table 12: Average Computational Times (in seconds) of SCSF under Different Degree Parameters $m$.

We investigated the impact of different degree parameters $m$ on the performance of SCSF. As shown in Table 12, the experiments were conducted on the Helmholtz operator dataset with a matrix dimension of 6400, a solution accuracy of 1e-8, 400 eigenvalues to be solved, and an inherited subspace size of 80. The degree parameter $m$, as described in Algorithm 3, primarily controls the order of the Chebyshev polynomial. The results indicate that varying $m$ within the range of 12 to 40 has a minimal effect on the computation time of SCSF. Therefore, as long as $m$ is chosen within a reasonable range, its specific value does not significantly influence the performance. In the main experiments of this paper, $m$ is fixed at 20.

### D.4.2  Subspace Dimension

| Dim | 50 | 60 | 70 | 80 | 90 | 100 | 110 | 120 |
|---|---|---|---|---|---|---|---|---|
| Time (s) | 43.28 | 44.35 | 42.43 | 40.52 | 39.65 | 37.43 | 38.28 | 38.58 |

Table 13: Average Computational Times (in seconds) of SCSF under Different Subspace Dimension.

We examine the influence of different inherited subspace sizes on the performance of SCSF. As presented in Table 13, the experiments are conducted on the Helmholtz operator dataset with a matrix dimension of 6400, a solution accuracy of 1e-8, 400 eigenvalues to be computed, and a degree parameter $m$ set to 20.

The results demonstrate that as the inherited subspace size increases, the computation time of SCSF initially decreases and then rises, reaching its minimum around a size of 100. The reduction in computation time at the front end is attributed to the enriched initial subspace with more available information as the inherited subspace grows. Conversely, the increase in computation time at the back end is due to the significantly higher overhead of performing Chebyshev filtering with a larger inherited subspace.

Overall, as long as the inherited subspace size is set within a reasonable range, its impact on SCSF remains minimal. In our experiments, we consistently set the inherited subspace size to 20% of the number of eigenvalues to be computed.

## D.4.3 Truncation Threshold for Low Frequencies

|                        | No sort | $p_0 = 10$ | $p_0 = 20$ | $p_0 = 30$ | $p_0 = 40$ | Greedy |
|------------------------|---------|------------|------------|------------|------------|--------|
| One-sided distance     | 0.95    | 0.89       | 0.85       | 0.85       | 0.85       | 0.85   |
| Sort time (s)          | 0       | 110        | 151        | 193        | 246        | 593    |
| Average solve time (s) | 66.7    | 52.2       | 40.5       | 40.5       | 40.5       | 40.5   |

Table 14: Average Computational Times (in seconds) of SCSF under Different Truncation Thresholds.

We measure the similarity between matrices by computing the cosine of the principal angles between their 10-dimensional invariant subspaces (spanned by the smallest 10 eigenvectors in modulus) (one-sided distance). Smaller values indicate higher similarity. As presented in Table 14, the experiments are conducted on the Helmholtz operator dataset with a matrix, a solution accuracy of 1e-8, 400 eigenvalues to be computed, and a degree parameter $m$ set to 20, 10k data problems, parameter matrix $P$ with dimension $p = 80$, and varying truncation frequencies $p_0$

The results demonstrate that sorting significantly increases inter-problem correlation in the dataset (explaining the performance gain). The truncation parameter $p_0$ affects sorting time, sorting quality, and solver time. For $p_0 \geq 20$, solver time becomes stable, showing diminishing returns. This reflects the interplay between sorting and Chebyshev iteration. In the main experiments of this paper, $p_0$ is fixed at 20.

