# OpenReview forum: "Accelerating Eigenvalue Dataset Generation via Chebyshev Subspace Filter"
_NeurIPS.cc/2025/Conference — Submitted to NeurIPS 2025_

### Official Review · Reviewer_GvsX · 2025-06-22

**Clarity:** 3
**Significance:** 2
**Originality:** 3
**Rating:** 4
**Confidence:** 3

**Summary:**

The authors propose a novel method called Sorting Chebyshev Subspace Filter (SCSF) to accelerate the generation of eigenvalue datasets for training neural eigenvalue methods. The key insight is that operators in datasets often share similarities since they describe related physical phenomena, but existing approaches solve each eigenvalue problem independently. SCSF addresses this by first using truncated FFT sorting to group operators with similar eigenvalue distributions, then employs Chebyshev filtered subspace iteration to leverage eigenpairs from previously solved problems when tackling subsequent ones. They claim this is the first method specifically designed to accelerate eigenvalue dataset generation and demonstrate up to 3.5× speedup compared to state-of-the-art solvers across four different operator types.

**Questions:**

I have several questions that I think could help improve the paper:

1. Can the authors provide better theoretical insight into when SCSF is expected to work well? Specifically, what properties of the operator distribution or dataset characteristics make SCSF most beneficial? This would help practitioners understand when to apply the method.
2. How sensitive is the method to the quality of the initial sorting? The authors show that sorting helps, but what happens when the similarity metric doesn't capture the true relationships between operators well? Have they considered adaptive or iterative sorting strategies?
3. The paper focuses on Hermitian matrices, but many practical eigenvalue problems involve non-Hermitian or generalized eigenvalue problems. How would SCSF extend to these cases, and what modifications would be needed?
4. Could the authors provide more analysis of the computational trade-offs? While they show speedups, a more detailed breakdown of when the sorting overhead is justified would be valuable. For instance, what's the break-even point in terms of dataset size?

**Ethical Concerns:**

["NO or VERY MINOR ethics concerns only"]

**Final Justification:**

The paper addresses a real practical bottleneck and provides a technically sound solution with solid experimental validation. I am glad the authors addressed all my concerns in detail. However, the work remains application-specific with incremental novelty rather than broad scientific impact and hence why i wont be increasing my score.

**Limitations:**

I think the authors have done a reasonable job addressing limitations, particularly in Section 5. They acknowledge that the method is currently focused on general eigenvalue problems and that extending to nonlinear eigenvalue problems would require additional work. They also mention the potential for better distance metrics in the sorting algorithm. However, I think they could expand the limitations discussion to include scenarios where the method might not work well, such as when operators in the dataset are highly dissimilar or when the dataset is small.

**Quality:**

3

**Strengths And Weaknesses:**

Strengths: I think the authors have tackled an important and practical problem. The computational cost of generating eigenvalue datasets is indeed a significant bottleneck for training neural eigenvalue methods, and the authors correctly identify that existing approaches ignore the inherent similarities between operators in these datasets. The technical approach seems sound and well-motivated. The authors cleverly combine two existing techniques - truncated FFT sorting and Chebyshev filtered subspace iteration - in a novel way that's specifically tailored for dataset generation. I appreciate that they provide detailed algorithmic descriptions and complexity analysis. The experimental evaluation is quite comprehensive, covering four different operator types (Poisson, elliptic, Helmholtz, vibration) and comparing against multiple baseline solvers.

Weaknesses: However, I have several concerns about the work. First, the theoretical analysis is somewhat limited. While the authors provide complexity analysis, they don't give much insight into when we can expect the method to work well versus when it might fail. The success of SCSF heavily depends on the similarity structure in the dataset, but there's limited discussion of how to characterize or predict this similarity. Second, I think the authors could better analyze the sorting overhead. While they show that truncated FFT sorting is faster than naive greedy sorting, the overall cost-benefit analysis could be more thorough. For small datasets or when operators are not very similar, the sorting overhead might outweigh the benefits. Third, the experimental section, while comprehensive in breadth, could use more depth in some areas. For instance, the authors don't thoroughly explore failure cases or discuss when their method performs poorly. The hyperparameter analysis is relegated to the appendix, but some of this seems important for understanding the method's sensitivity.

On the clarity front, the paper is generally well-written, though some sections feel dense. The method description is clear, but I think the authors could better motivate why Chebyshev filtering specifically is the right choice here versus other subspace iteration methods.
In terms of significance, this addresses a real practical problem, but the impact might be somewhat limited since it's specifically focused on eigenvalue dataset generation rather than eigenvalue solving more broadly. The originality is good - while the individual components exist, combining them for this specific application appears novel.

---

> ### Author Rebuttal · Authors · 2025-07-31
>
> Dear Reviewer,
>
> Thank you for your thoughtful and constructive review. Your feedback is immensely helpful for strengthening our paper, and we offer the following responses to address your concerns.
>
> ### **On Theoretical Insights, Applicability, and Failure Cases (W1, W3, Q1, Q2)**
>
> Thank you for these insightful points. The performance of SCSF hinges on the **continuous relationship** between operator parameters and the resulting matrix properties, a common characteristic in datasets for scientific ML where parameters are generated smoothly (e.g., via Gaussian Random Fields)[1-3]. In this setting, our sorting algorithm effectively groups similar problems, maximizing acceleration. Conversely, if the problems are independent or discontinuous, the sorting provides little benefit, and SCSF's performance gracefully degrades to that of a warm-started Chebyshev filter.
>
> To provide concrete insight, we conducted two supplementary experiments:
>
> **Experiment 1: Quantifying the Impact of Similarity.** We simulated datasets with controlled similarity by perturbing the parameters of a Helmholtz operator. The "perturbation size" indicates the degree of dissimilarity between consecutive problems.
>
> | Perturbation Size | Eigsh | LOBPCG | ChFSI | SCSF (w/o sort) | SCSF |
> | -- | -- | -- | -- | -- | -- |
> | 50% | 151 | 130  | 107 | 76  | 27 |
> | 10%  | 150 | 129 | 107 | 48 | 23 |
> | 1% | 152  | 130 | 107   | 14 | 6 |
> | 0% (Identical)    | 151   | 130  | 107   | 2  | 2  |
> | Standard Method   | 152   | 130 | 107   | 82  | 31   |
> | Independent Init. | 152   | 130  | 107  | 10 | 107  |
>
> *Note: This experiment was run on the Helmholtz dataset (dim=6400, L=200). Times are in seconds. The latter problem is obtained by perturbing the parameters of the former. The perturbation is obtained by multiplying the perturbation size by the Gaussian random field. The perturbation size can be considered a measure of the internal similarity of the dataset. The "standard  method" is the same as the experimental setup in the paper. "Independent Init." can be considered a special case where the operator parameters and the resulting matrix are completely independent and discontinuous.*
>
> This shows: 1) Performance is strongly correlated with problem similarity. 2) Our sorting algorithm is highly effective at enhancing this correlation (SCSF vs. SCSF w/o sort). 3) In the worst case of completely independent problems, SCSF degrades to ChFSI, as expected.
>
> **Experiment 2: Simulating a Discontinuous Failure Case.** We mixed two different datasets (Helmholtz and Poisson, 1:1 ratio) to simulate a discontinuous problem sequence.
>
> |  | Eigsh | LOBPCG | ChFSI | SCSF (w/o sort) | SCSF |
> | -- | -- | -- | -- | -- | -- |
> | Time (s) | 154   | 280    | 132   | 118  | 98   |
>
> This shows that SCSF's performance gain is reduced when correlation is disrupted, and the benefit of sorting also decreases, aligning with our expectations.
>
> **Regarding adaptive sorting strategies:** This is an excellent suggestion. While not explored in the current work, we envision future extensions where the solving order could be dynamically adjusted based on runtime information. This would be particularly useful for online settings where data is acquired sequentially (e.g., from real-time sampling or iterative data generation guided by model performance). This is a promising direction for our future research.
>
> **Action:** We will incorporate these experiments and a more detailed discussion on applicability and failure modes into Sections 3 and 5 of the revised manuscript. We will also expand the limitations section to explicitly state that SCSF is designed for datasets with underlying continuity.
>
> ### **On the Overhead and Cost-Benefit of Sorting (W2 & Q4)**
>
> Thank you for raising this important point about the computational trade-offs.
>
> - **Sorting Overhead Analysis:** The overhead of our sorting algorithm is minimal in practice. As shown in **Table 4**, sorting a large dataset of 10^4 samples takes approximately 151 seconds. In contrast, solving a single eigenvalue problem on the Helmholtz dataset (400 eigenvalues) takes an average of 253 seconds with Eigsh. For a dataset of this size, the total sorting overhead constitutes **less than 0.1%** of the total generation time, making its cost negligible.
> - **Benefit and Break-Even Point:** The benefit of sorting is a significant reduction in the number of solver iterations and total floating-point operations (Flops). **Table 3** shows that sorting reduces iterations by 5-50% and total Flops by 7-43%. Given that the sorting cost is a tiny fraction of the total computation, a "break-even point" is not a practical concern for typical dataset sizes. The more relevant goal is to find the *fastest sorting algorithm that still effectively captures the similarity structure*. Our Truncated FFT Sort achieves this, providing nearly identical solver performance as the far more expensive greedy sort (**Table 5**), demonstrating its cost-effectiveness.
>
> **Action:** We will add a more detailed analysis, potentially with a plot illustrating the relationship between sorting overhead and total time saved, to the appendix of the revised paper.
>
> ### **On Experimental Depth and Hyperparameter Analysis (W3)**
>
> We appreciate your feedback on making the hyperparameter analysis more accessible. We conducted a detailed analysis of key hyperparameters (filter degree `m`, subspace dimension, and truncation threshold `p_0`) in **Appendix D.4**. Our main finding is that **SCSF is robust to these hyperparameters within reasonable ranges.** For instance, varying the filter degree `m` between 16 and 40 has a minimal effect on computation time , and setting the truncation threshold `p_0` to a small value (e.g., 20) achieves solver performance nearly identical to that of the costly full greedy sort.
>
> **Action:** To improve the main paper's clarity, we will move a concise summary of these key findings on hyperparameter robustness from the appendix into the main experimental section, while keeping the detailed tables and analysis in the appendix for interested readers.
>
> ### **On Motivation for Method Choice and Significance (W4 the clarity front)**
>
> Thank you for these two key points.
>
> - **Why Chebyshev Filtering?** Our choice was guided by the empirical results presented in our ablation study in **Section 4.3**. As shown in **Table 2**, when we modified other standard solvers (Eigsh, LOBPCG, KS, JD) to use a "warm start" from the previous solution, only LOBPCG saw a notable speedup, while SCSF (which uses Chebyshev filtering) was significantly faster than all of them. The mechanism behind this is that the Chebyshev filter, via a polynomial transformation, is exceptionally effective at amplifying the components of the target invariant subspace while suppressing others. This makes it uniquely suited to capitalize on a high-quality initial subspace provided by a similar, previously solved problem.
>   - **Action:** We will add a more explicit motivation for this choice in Section 3.2.
> - **Why Focus on Dataset Generation?** Our core contribution is not a new general-purpose eigensolver, but rather an efficient and lossless optimization of traditional solvers for the specific, emerging problem of dataset generation in AI. This addresses a widely recognized bottleneck in "AI for Science." As a leading survey notes [4] P169, "A primary limitation of learned solvers is the requirement of an adequate number of training data generated by costly numerical solvers". An efficient generation tool lowers the barrier to entry and accelerates the research cycle for the entire community. Furthermore, the core idea—transforming a batch of related problems into a "sequential problem" to be solved with warm starts—has broader potential in fields like optimization, signal processing, and stability analysis, which we plan to explore in future work.
>
> ### **On Extending SCSF to Other Eigenvalue Problems (Q3)**
>
> This is an excellent, forward-looking question. Extending SCSF to non-Hermitian, generalized, or nonlinear eigenvalue problems is an important direction for future work. Our current thinking involves:
>
> 1. **Adapting the Solver:** We would need to employ or develop Chebyshev-like iterative methods suitable for these other problem types. Versions for generalized eigenvalue problems already exist. For nonlinear problems, a common approach is to linearize them into a sequence of linear eigenvalue problems, a process where the SCSF paradigm could be applied to accelerate the solution of the sequence.
> 2. **Designing New Sorting Metrics:** Our current sorting relies on the distance between parameter matrices. This may not be optimal for all problem types. New, specially designed metrics would likely be needed to effectively capture problem similarity in these different contexts.
>
> **Action:** Based on your suggestion, we will expand this discussion in the revised version to also include the potential for extension to non-Hermitian and generalized eigenvalue problems, providing a more comprehensive view of our work's current scope and future directions.
>
> **References:**
>
> [1] Fourier neural operator for parametric partial differential equations, ICLR 2019.
>
> [2] Neural krylov iteration for accelerating linear system solving, NIPS 2024.
>
> [3] Accelerating data generation for neural operators via krylov subspace recycling, ICLR 2024.
>
> [4] Artificial Intelligence for Science in Quantum, Atomistic, and Continuum Systems, 2023.
>
> ---
>
> **Thanks again**
>
> We sincerely thank you again for your constructive and detailed feedback, which has helped us identify clear pathways to improve our paper. Should you have any further questions or require additional discussion, please don't hesitate to reach out. If we have adequately addressed your concerns, we would be grateful for your consideration in adjusting your evaluation score accordingly.

---

> > ### Comment · Reviewer_GvsX · 2025-08-06
> > **Response to authors**
> >
> > Thanks for the detailed answers! I am happy with the response.

---

> ### Author Response · Authors · 2025-08-06
> **Thanks for Your Feedback and Revision Plan**
>
> We greatly appreciate your constructive feedback! We are pleased to hear that our responses have resolved most of your concerns. As suggested, we will incorporate the newly added results into the final version of the paper.

---

### Official Review · Reviewer_QhCL · 2025-06-30

**Clarity:** 4
**Significance:** 3
**Originality:** 3
**Rating:** 3
**Confidence:** 2

**Summary:**

This paper provides an algorithm that speedup the eigenvalue computation for a sequence of matrices that all originated from the same partial differential equations. It has two contributions 1)  To benefit the successive solving sequence of the eigenvalue problem, one way is to sort the matrices that pull matrices with similar spectral properties close in the solving sequence. The solution of one matrix can then be used to provide a warm start to the solution of the next matrix in the sequence. Typically, this can be done with the F-norm. However, since the eigenvalues are dependent on low-frequency info, the paper first uses truncated FFT to preprocess the matrices to retain only low-frequency information, then applies the sorting using the F-norm. 2) It proposed a method that can do a warm start well, based on Chebyshev filtered subspace iteration.

**Questions:**

None

**Ethical Concerns:**

["NO or VERY MINOR ethics concerns only"]

**Limitations:**

Yes

**Quality:**

3

**Strengths And Weaknesses:**

Strength

- The paper is well written and easy to understand
- The proposed approach makes intuitive sense
- The results do show that there is a speedup of 3.5x overall compared with other methods.

Weakness:
- I wonder if NeurIPS is the best venue for this paper? I can understand that the motivation for developing this approach is to speed up the generation of benchmark matrices and the computation of the labels (eigenvalues). But a conference in the numerical analysis field could provide a better gauge as to the significance of the proposed methods

- A factor of 3.5x is significant in general. But if benchmark generation is a one-off, I wonder if it is that important, or if we can afford to wait for the results. Perhaps solving the eigenvalues of a sequence of matrices has other applications that make this contribution more significant? Or perhaps the methods, esp Chebyshev filtered subspace iteration, can motivate improvement to other fundamental numerical analysis methods (see my question as to whether a numerical analysis conference is a better fit)

---

> ### Author Rebuttal · Authors · 2025-07-31
>
> Dear Reviewer,
>
> We sincerely thank you for your positive feedback on our paper's clarity and the intuitive nature of our approach. We are grateful for your thoughtful comments, which give us the opportunity to further clarify the significance and positioning of our work. We hope our response below will address your concerns and persuade you of our paper's value to the NeurIPS community.
>
> ---
>
> ### **On the** **Fit** **of Our Work for NeurIPS (W1)**
>
> We greatly appreciate your perspective on the most suitable venue for our work. While our SCSF method is indeed rooted in numerical analysis techniques, we wish to emphasize that its **motivation, application, and intended audience are firmly centered within the** **machine learning** **community, particularly the rapidly growing field of "****AI** **for Science."**
>
> 1. **Motivation Stemming from the** **AI** **Community:** Our primary motivation, as you correctly noted, is to accelerate the generation of labeled datasets required to train data-driven models like [1] ICLR 2019, [2] NIPS 2024 spotlight. NeurIPS has become a premier platform for showcasing such cutting-edge AI for Science research. Our work directly addresses a critical bottleneck faced by this community, aiming to provide a foundational tool that enables more efficient research.
> 2. **Serving the Needs of the AI Community:** The challenge we address is widely recognized within the field.
>    1. As noted by Prof. Max Welling et al. in their ICML 2022 paper [3], the AI community faces a "proverbial ‘chicken-and-egg problem’": training neural solvers requires abundant data that must first be generated by classical solvers.
>    2. A comprehensive survey on AI for Science [4] (P169) highlights this directly: "A primary limitation of learned solvers is the requirement of an adequate number of training data generated by costly numerical solvers."
>    3. Our research aims to lower this barrier to entry. Currently, generating a large-scale eigenvalue dataset can take days or even weeks. By dramatically accelerating this process, our method enables researchers, especially those with limited access to large-scale computing resources, to participate more actively in this field, thereby fostering innovation across the entire AI for Science ecosystem.
> 3. **Bridging** **Two Fields:** We believe that innovatively applying established numerical analysis techniques (like Chebyshev filtering) to solve practical problems in machine learning is precisely the kind of interdisciplinary research that NeurIPS champions. Our work not only provides a tool for the AI community but also showcases a novel application scenario for numerical analysis techniques in the AI era, serving as a valuable bridge between the two domains.
> 4. **Precedent at Top** **AI** **Venues:** There is a strong precedent for work on accelerating dataset generation at top AI conferences. Recent papers at ICLR 2024 spotlight [5] and ICML 2024 [6] have focused on accelerating the solution of linear systems for this purpose. Our paper builds on this emerging topic, and to our knowledge, is the **first to specifically address the acceleration of** ***eigenvalue*** **dataset generation**, a distinct and equally important challenge.
>
> For these reasons, we are confident that NeurIPS is the ideal platform to present this work, as it directly confronts a pressing issue faced by the conference's core audience.
>
> ### **On the Significance of the 3.5x Speedup (W2)**
>
> Thank you for this insightful question regarding the importance of our method's performance gains. We would like to elaborate on the significance of our work from three perspectives.
>
> 1. **The 3.5x Speedup is a Conservative Estimate:** The "3.5x" figure mentioned in the abstract is a conservative summary that holds true across all tested scenarios. In many specific cases, the performance gains were far more substantial. For instance, on the Helmholtz dataset, SCSF achieved speedups of up to **8x, 20x, 6x, and even 95x** compared to various state-of-the-art solvers. Reducing computation time from weeks to days represents a transformative improvement in research efficiency and iteration speed.
> 2. **Dataset Generation is Not a "One-Off" Task:** In the practice of AI for Science, data generation is a recurring and essential part of the research lifecycle.
>    1. **Diverse Scientific Problems:** Different physical phenomena require unique, customized datasets. The variety of possible differential operators is immense, meaning there will always be a need to generate new data for new problems. This is a primary driver for research in data generation and augmentation.
>    2. **Iterative Research and Exploration:** Scientific inquiry is an iterative process. Researchers frequently adjust parameter ranges, discretization precision, or operator forms to explore physical laws under different conditions. Each adjustment typically requires generating an entirely new dataset.
>    3. Model and Hyperparameter Tuning: Even for a single problem, the process of finding an optimal model architecture and its hyperparameters often involves evaluation on datasets of varying scales or distributions.
>    4. Therefore, an efficient data generation pipeline is a critical enabler for the entire iterative AI for Science research paradigm.
> 3. **Broader Application Horizons:** You have keenly pointed out the potential for sequential eigenvalue solving in other domains. Beyond dataset generation, this task is crucial in numerous scientific and engineering fields where sequences of similar matrices arise. Potential applications include:
>    1. **Data analysis** **in** **machine learning** **and** **optimization**, where sequences of closely related matrices are often analyzed.
>    2. **Signal processing in communications and stability analysis in power systems**, which frequently involve solving for eigenvalues of highly similar matrices.
>
> In our future work, we plan to adapt the core ideas of SCSF to these domains, transforming large sets of eigenvalue problems into sequential tasks to be solved efficiently.
>
> In summary, our work provides a significant performance improvement for a critical and recurring task in AI for Science, and its core methodology holds promise for empowering a much broader range of scientific and engineering applications.
>
> **References:**
>
> [1] Fourier neural operator for parametric partial differential equations, ICLR 2019.
>
> [2] Neural krylov iteration for accelerating linear system solving, NIPS 2024.
>
> [3] Lie Point Symmetry Data Augmentation for Neural PDE Solvers, ICML 2022.
>
> [4] Artificial Intelligence for Science in Quantum, Atomistic, and Continuum Systems, 2023.
>
> [5] Accelerating data generation for neural operators via krylov subspace recycling, ICLR 2024.
>
> [6] Accelerating pde data generation via differential operator action in solution space, ICML 2024.
>
> ---
>
> #### **Thanks again**
>
> We sincerely thank you again for your constructive and detailed feedback, which has helped us identify clear pathways to improve our paper. Should you have any further questions or require additional discussion, please don't hesitate to reach out. If we have adequately addressed your concerns, we would be grateful for your consideration in adjusting your evaluation score accordingly.

---

> ### Author Response · Authors · 2025-08-05
> **To Reviewer QhCL: Thanks and Follow-up Discussion**
>
> Dear Reviewer QhCL,
>
> Thank you very much for taking the time to read our rebuttal and for promptly acknowledging it. We truly appreciate your engagement with our work during the review process.
>
> If you have any further thoughts, questions, or suggestions---either regarding our current response or the broader direction of the work---we would be genuinely grateful to hear them. We highly value your perspective and would welcome any opportunity for continued discussion or clarification.
>
> Best,
>
> Authors

---

> ### Author Response · Authors · 2025-08-07
> **To Reviewer QhCL: Thanks and Follow-up Discussion**
>
> Dear Reviewer QhCL,
>
> Thank you again for your thoughtful and constructive review.
>
> Following up on our submitted rebuttal, we would be very grateful to hear your thoughts, especially as the discussion period is nearing its end. Your feedback is crucial for us to understand if our responses have adequately addressed your concerns and to help us further improve our paper.
>
> We are happy to provide any further clarifications and look forward to discussing this with you.
>
> Best regards,
>
> Authors

---

### Official Review · Reviewer_snRn · 2025-07-01

**Clarity:** 3
**Significance:** 2
**Originality:** 2
**Rating:** 4
**Confidence:** 3

**Summary:**

This paper considers the modern "parametric operator learning framework" such as DeepONet, which tries to directly learn a mapping from an operator to its eigenvalue pairs. This is a supervised learning framework that requires such a "labeled" dataset {(operator, eigenvalue pairs)}, and this work aims to reduce the time to generate the labeled dataset. The parametric operator framework works upon the assumption that operators in a dataset have enough commonalities to be exploited, so, for example, they assume a class of PDEs with varying parameters. In this setting, they want to expedite to get the eigenvalue pairs for different parameters. Their methodology is essentially as follows:
1) Sort the parameters by applying "FFT" to the parameters.
2) Given the sorted parameters, they sequentially solve each eigenvalue problem, but utilizing the previous iteration via the "Chebyshev filtered subspace iteration."
The resulting algorithm is evaluated on four different eigenvalue problems, and the results compared to several existing methods.

**Questions:**

1.  To the extent that it differs from what is written above, could the authors summarize what they would characterize as most novel in the work's methodology?

2.  Does the proposed method have any concrete optimality properties?  To what extent does it bound achievable speeds?

3.  How were the datasets chosen for the experiments?  The choices seem a bit arbitrary.   Are they representative in some way?  Are there examples of operator eigenvalue problems that create particular challenges for the methodology?

**Ethical Concerns:**

["NO or VERY MINOR ethics concerns only"]

**Final Justification:**

I conclude that this is a problem of the community's interest, the proposed solution is simple and intuitive, and that it appears to be empirically effective. Whether a supervised approach is ultimately the right way to go, I'm not sure. So longer term value unclear.

**Limitations:**

Yes.

**Paper Formatting Concerns:**

None.

**Quality:**

3

**Strengths And Weaknesses:**

Strengths:

- the problem is one of growing interest in a range of "AI for science" problems

- it highlights the role that the established Chebyshev subspace filtering methods can play in efficiently solving eigenvalue problems at scale

- the empirical results demonstrate the computational complexity can be significantly smaller than several existing methods.

Weaknesses:

- the novelty of the paper is rather limited.   In particular, the addition to the methodology beyond the standard Chebyshev filtering appears to be introduction of the "parameter FFT", which while useful is a somewhat incremental contribution.

- generally the writing is good, but the transition to Section 2.2 is rather abrupt...the motivation and context are absent, and is disconnected from what precedes it.  Moreover, the writing is too technical to appreciate at the point it appears.

---

> ### Author Rebuttal · Authors · 2025-07-31
>
> Dear Reviewer,
>
> Thank you for your constructive feedback and for recognizing our work's relevance and empirical performance. Your insightful questions have helped us clarify our contributions. We hope our responses below adequately address your concerns and will encourage you to reconsider your score.
>
> ---
>
> ### **On the Novelty of Our Work (W1 & Q1)**
>
> Our core innovation is not merely combining existing techniques, but the **first systematic formulation and solution for accelerating operator eigenvalue dataset generation.**
>
> Our novelty is threefold:
>
> 1. **Problem Formulation and Framework:** Traditional methods treat each eigenvalue problem in a dataset as an isolated task, leading to significant computational redundancy. We are the first to reframe "eigenvalue dataset generation" as a "sequential eigenvalue solving" problem. Based on this new perspective, we propose a novel **"Sort-Solve" framework** designed to maximally exploit the intrinsic similarities among operators.
> 2. **Systematic Methodological Integration:** As shown in our experiments (Sec 4.4, Page 9, Line 284), ChFSI performs poorly on unsorted, disordered problems, sometimes even slower than standard solvers. Likewise, sorting is of little value without an efficient solver that can leverage the created sequence. Our key contribution lies in the **systematic integration** of an efficient "Truncated FFT Sort" with a "Chebyshev filter" optimized for sequential problems. The Truncated FFT Sort (Algorithm 2) enhances the correlation within the problem sequence at a very low computational cost (far cheaper than greedy sorting), creating the ideal conditions for the Chebyshev filter to thrive. This creates a synergistic effect where the whole is greater than the sum of its parts.
> 3. **Significant Empirical Value:** As shown in our ablation study (Sec 4.3, Page 8, Line 246), we experimented with combining sorting with various traditional algorithms and found that the combination in SCSF delivered the most stable and significant performance gains. SCSF achieves state-of-the-art performance across four distinct datasets, with speedups ranging from **3.5x to 95x**. This demonstrates the effectiveness of our proposed framework and provides a practical, valuable tool for the AI for Science community to overcome the data generation bottleneck.
>
> Therefore, our novelty lies not in a single component, but in the new problem formulation, the systematic design of our integrated method, and the exceptional performance it achieves.
>
> To directly answer your question, **we consider the "Truncated FFT Sort" (Algorithm 2) to be the most novel component of our methodology.** Its ingenuity is rooted in a key insight: the spectral properties of operators are primarily influenced by the **low-frequency components** of their parameters. By performing an FFT on the parameter matrices and sorting based only on these retained low-frequency components, we achieve:
>
> - **High Efficiency:** The computational complexity of FFT is O(p^2logp), far more efficient than a greedy algorithm that directly computes distances on high-dimensional parameter matrices. As shown in Table 4, for a dataset of size 104, our sorting algorithm is nearly **4 times faster** than greedy sorting.
> - **Excellent Effectiveness:** As shown in Table 5, the final solution efficiency achieved using our low-cost sorting method is **nearly identical** to that achieved using the computationally expensive greedy sort.
>
> This "low-cost, high-yield" sorting strategy is the key enabler for our SCSF framework and represents a significant and novel methodological contribution.
>
> ### **On Writing and Structure (W2)**
>
> Thank you for the valuable suggestion. We agree Section 2.2 was abrupt. In the revision, we will add a transitional paragraph to motivate the need for Chebyshev polynomial preliminaries and move most of the technical derivations to the appendix, keeping only the core definitions in the main text to improve flow.
>
> ### **On Optimality Properties and Speed Bounds (Q2)**
>
> This is a very insightful question. From a theoretical standpoint, our method does not currently possess a formal proof of "optimality," and its performance upper bound is difficult to define with a single formula, as it is highly dependent on the intrinsic properties of the dataset.
>
> - **On Optimality:** The performance of SCSF is strongly correlated with the "similarity" among operators in the dataset. In an ideal scenario where two consecutive matrices, A(i−1) and A(i), are nearly identical, using the eigenvectors of A(i−1) as the initial subspace for A(i) could lead to convergence in just one or two Chebyshev filter iterations. Our sorting algorithm is a heuristic approach designed to approximate an optimal sequence that maximizes this similarity.
> - **On Speed Bounds:** The upper limit of the speedup depends on two factors: (1) **Post-sorting similarity:** Higher similarity leads to fewer iterations and greater acceleration potential. (2) **Filter** **operation overhead:** As shown in Table 11, the Chebyshev filter step itself consumes over 70% of SCSF's total computation time. Therefore, even if the number of iterations were reduced to a minimum (e.g., 1), the total speedup is bounded by the fixed cost of this single iteration.
>
> To better illustrate the theoretical performance ceiling, we conducted a simulation. We generated a sequence of Helmholtz operator problems where each subsequent problem is a slight perturbation of the previous one. The magnitude of this perturbation reflects the dataset's internal similarity. The experiment was run on the Helmholtz dataset (dim=6400, L=200). Average solution times (in seconds) are reported.
>
> | Perturbation Size    | Eigsh | LOBPCG | ChFSI | SCSF (w/o sort) | SCSF |
> | --- | --- | --- | --- | --- | --- |
> | 50%   | 151   | 130    | 107   | 76  | 27 |
> | 10%   | 150   | 129    | 107   | 48 | 23 |
> | 1%  | 152   | 130    | 107   | 14  | 6    |
> | 0% (Identical)    | 151   | 130    | 107   | 2  | 2  |
> | Standard Generation  | 152   | 130    | 107   | 82 | 31   |
> | Independent Problems | 152   | 130    | 107   | 107    | 107  |
>
> *Standard Generation* refers to the setup in our paper. *Independent Problems* simulates a case with no exploitable similarity, where each solver is initialized independently.
>
> This experiment shows:
>
> 1. The theoretical performance limit is approximately **2 seconds**, corresponding to 1-2 subspace iterations.
> 2. Performance is heavily influenced by dataset correlation. The more related the problems are, the greater the advantage of SCSF and ChFSI.
> 3. The sorting algorithm in SCSF is highly effective at enhancing problem correlation, as the performance of SCSF is consistently better than SCSF w/o sort until the problems are already highly correlated.
> 4. If the problems are completely independent, SCSF's performance degrades to that of ChFSI, as expected.
>
> **Action:** In the final version, we will incorporate this experiment and analysis to provide a clearer picture of our method's performance bounds.
>
> ### **On the Choice of Experimental Datasets (Q3)**
>
> Our choice of four datasets was deliberate, aiming for comprehensive evaluation.
>
> **Representativeness:**
>
> The datasets cover key differential operators in science and engineering (Poisson, Elliptic, Helmholtz, Vibration). These operators vary in their mathematical properties (e.g., order) and physical origins, providing a robust testbed for our method's universality. Furthermore, these operators and their corresponding PDE problems are commonly used as benchmarks in mainstream neural operator papers [1, 2, 3], ensuring the relevance of our work to the community.
>
> **Challenging Problems:**
>
>  Our method assumes a continuous relationship between operator parameters and matrix properties, which holds for most data generation scenarios. If this assumption fails (e.g., the problems are discontinuous), the sorting module becomes ineffective, and SCSF performance degrades to the "w/o sort" version. Even so, as shown in Table 3, the "w/o sort" SCSF is still 1.2 to 1.5 times faster than a randomly initialized ChFSI, demonstrating that simply using the previous solution as an initial guess provides a benefit.
>
> To further illustrate this, we simulated an extreme case by mixing the Helmholtz and Poisson datasets (1:1 ratio, dim=6400, L=200), creating a discontinuous sequence.
>
> |          | Eigsh | LOBPCG | ChFSI | SCSF (w/o sort) | SCSF |
> | -------- | ----- | ------ | ----- | --------------- | ---- |
> | Time (s) | 154   | 280    | 132   | 118             | 98   |
>
> The results confirm our expectations:
>
> 1. SCSF's performance gain is reduced in this discontinuous scenario because the internal correlation is disrupted.
> 2. The effectiveness of sorting is diminished, as the parameters no longer reliably predict matrix similarity across dataset types.
>
> In summary, SCSF is most effective in the common scenario where dataset generation involves continuous parameter variation.
>
> **Action:** We will add this experiment and analysis to the final version. We will also explicitly state this assumption in the limitations section and recommend that users assess the continuity of their data generation process before applying SCSF.
>
> **References:**
>
> [1] Fourier neural operator for parametric partial differential equations, ICLR 2019.
>
> [2] Neural krylov iteration for accelerating linear system solving, NIPS 2024.
>
> [3] Accelerating data generation for neural operators via krylov subspace recycling, ICLR 2024.
>
> ----
>
> **Thanks again**
>
> We sincerely thank you again for your constructive and detailed feedback, which has helped us identify clear pathways to improve our paper. Should you have any further questions or require additional discussion, please don't hesitate to reach out. If we have adequately addressed your concerns, we would be grateful for your consideration in adjusting your evaluation score accordingly.

---

> > ### Comment · Reviewer_snRn · 2025-08-05
> > **comments on response**
> >
> > Thank you for the further comments, which are helpful. I have also read the other reviews and their responses.  I conclude that this is a problem of the community's interest, the proposed solution  is simple and intuitive, and that it appears to be empirically effective.  Whether a supervised approach is ultimately the right way to go, I'm not sure.  So longer term value unclear.  But I am willing to set this concern aside and raise my score from 3 to 4 at this point.

---

> ### Author Response · Authors · 2025-08-05
> **Thanks for Your Feedback and Revision Plan**
>
> We greatly appreciate your constructive feedback and your willingness to raise your scores! We are pleased to hear that our responses have resolved most of your concerns. As suggested, we will incorporate the newly added results into the final version of the paper.

---

### Official Review · Reviewer_Hx22 · 2025-07-10

**Clarity:** 2
**Significance:** 2
**Originality:** 3
**Rating:** 4
**Confidence:** 4

**Summary:**

This paper introduced a new approach to generating eigenpairs for supporting data-driven eigenvalue methods. The main method is based on a Chebyshev subspace filter and makes use of the structures of operators. The authors presented numerical experiments to demonstrate the effectiveness of the proposed algorithm.

**Questions:**

1. How is the boundary condition dealt with in discretizing the operator problem to the matrix problem?
2. Could the authors comment on how the parameters in FDM affect the error of computed eigenpairs?
### Comments:
+ Eq. (2) is not a good example for a 2D problem; at least $3 \times 3$ grids are needed to see how the discretization works.
+ L86, missing a space after the second comma;
+ Sec 2.2 is mostly facts; the authors might consider moving some parts to the supplementary.

**Ethical Concerns:**

["NO or VERY MINOR ethics concerns only"]

**Final Justification:**

The authors have addressed my concerns. The added table and illustrations also provide a boost to the current manuscript. Though it is still debatable whether the supervised regime is superior to the unsupervised one, the current results already provide useful information for practitioners. Based on these, I updated my ratings.

**Limitations:**

The intrinsic error and scalability issues introduced by FEM or similar discretization processes shall be included as a limitation of applicability.

**Quality:**

3

**Strengths And Weaknesses:**

### Strength
Generally, the technical writing is easy to follow; the overall analyses are technically sound. Experiment setups are reasonable and give positive results.

### Weakness
1. [Significance] It is unclear to me whether the dataset generation is the bottleneck for designing data-driven eigensolvers. Particularly, there have been recent works on solving the eigen/SVD problems directly from the operator itself, without requiring eigenpairs in its training. Examples include:
- Pfau, David, et al. "Spectral inference networks: Unifying deep and spectral learning." arXiv preprint arXiv:1806.02215 (2018).
- Deng, Zhijie, Jiaxin Shi, and Jun Zhu. "Neuralef: Deconstructing kernels by deep neural networks." International Conference on Machine Learning. PMLR, 2022.
- Ryu, J. Jon, et al. "Operator SVD with neural networks via nested low-rank approximation." arXiv preprint arXiv:2402.03655 (2024).

In comparison, this paper primarily focuses on classical numerical methods as baselines, which may be less relevant given current deep learning practices.

2. [Methodology] Since the proposed method relies on some existing numerical solvers, it is not guaranteed that the generated dataset can be used as a ground truth. In particular, matrix-based approaches are sensitive to the discretization process and can hardly be scaled to high-dimensional problems.
3. [Performance] Without context, it is not obvious to see whether a 3.5x speedup is a significant performance gain. Specifically, parallization could provide similar gain for the particular dataset generation problem.

---

> ### Author Rebuttal · Authors · 2025-07-31
>
> Dear Reviewer,
>
> Thank you for your detailed and insightful feedback. We have carefully considered your comments and provide our responses below, which we hope will clarify our contributions and address your concerns.
>
> ---
>
> ### **On the Significance of Accelerating Dataset Generation (W1)**
>
> Thank you for the valuable references to unsupervised methods [1, 2, 3]. While these represent an important research direction, our work focuses on the dominant **supervised** **paradigm** in scientific computing. Methods like FNO [4] and NeurKItt [5] require large, high-precision labeled datasets, and the significant computational cost of generating this data is a primary bottleneck. Our goal is to provide a crucial acceleration tool for these widely used methods. With speedups of 3.5x and often significantly higher, SCSF can reduce data generation cycles from weeks to days, greatly improving research efficiency.
>
> The two approaches have different scopes. Unsupervised methods are often slow and less accurate for the common low-dimensional problems found in engineering. Our supplementary experiment on a 2D Helmholtz problem illustrates this trade-off (solving for the first 100 eigenvalues, matrix dimension 6400).
>
> | Category        | Algorithm          | Inference Time | Training Time | Dataset Gen. Time | Relative Residual |
> | ------- | -------- | ------ | ---- | --------- | -------- |
> | Our Method      | SCSF (random init) | 1 min          | -             | -                 | 1e−8              |
> | Traditional     | Eigsh              | 1 min          | -             | -                 | 1e−8              |
> | Supervised NN   | NeurKItt           | 0.1s           | 3h            | 20h               | 1e−2              |
> | Unsupervised NN | NeuralEF           | 2h             | -             | -                 | 1e−2              |
> | NeuralSVD       | 3h                 | -              | -             | 1e−2              |                   |
>
> *Note: As SPIN is an earlier, slower algorithm, we compared against its improved versions, NeuralEF and NeuralSVD, using their official code. For a single problem, SCSF's initial* *vector* *is randomly initialized.*
>
> This experiment demonstrates that each approach has its place. Unsupervised methods avoid the need for data but are slow and less accurate for these problems. Supervised methods are the fastest at inference but require significant training and data generation time. This validates the importance of developing supervised methods and, consequently, the significance of accelerating their dataset generation.
>
> **Action:** In the final version, we will add a discussion of unsupervised methods to our "Related Work" section to better frame our contribution.
>
> ###  **On Methodology: "Ground Truth" and Scalability (W2)**
>
> Thank you for your sharp insights. We wish to clarify the reliability and scalability of our method.
>
> - **On the Reliability of Datasets as "Ground Truth":**
>   - In the machine learning context, "Ground Truth" refers not to an unobtainable analytical solution, but to a reference solution with precision far exceeding the predictive capability of the neural model being trained. Our data generation pipeline involves two steps: 1) operator discretization and 2) matrix eigenvalue solving. Our paper's core contribution is accelerating the second step, which is the computational bottleneck, often consuming over 95% of the total time.
>   - In all our experiments, the relative residual for the matrix solve (for both baselines and SCSF) is set to a high precision of **1e−8 to 1e−12**. Our tests confirm that the discretization error for all experiments is well below **1e−6**. This level of accuracy is orders of magnitude higher than the typical error range of neural network like FNO [4] or NeurKItt [5], which is usually between **10−3 and 10−5**. Therefore, the datasets generated with our method are a highly reliable "Ground Truth" for training such neural networks.
> - **On** **Discretization** **Sensitivity and Scalability:**
>   - We fully agree that the final accuracy and scale of matrix-based methods are tied to the initial discretization. However, our focus is not on the discretization technique itself, but on efficiently solving the resulting matrix problems.
>   - **Discretization-Agnostic:** SCSF method is agnostic to the choice of discretization. Whether a spectral method is used, SCSF can accelerate the process as long as it results in a sequence of similar matrix eigenvalue problems.
>   - **Standard and Relevant** **Discretization****:** The discretization methods and code used in our experiments are adopted directly from the benchmark dataset generation procedures of leading works like [4], [5], [6]. They use the exact same procedures to generate their "Ground Truth" training and test sets, ensuring our work is directly relevant and compatible with current best practices.
>   - **Performance on High-Dimensional Problems:** Your point about high-dimensional PDEs is crucial. These problems result in matrices with very large dimensions, making them challenging for traditional solvers. This is precisely where SCSF provides the most value. Our results in **Figure 3 and Table 10** clearly show that **SCSF's performance advantage grows as the matrix dimension increases**. This demonstrates that our method is even more critical for tackling the large-scale, complex problems that are most relevant to practical applications, helping to push past existing computational barriers.
>
> To further solidify this point, we trained NeurKItt [5] using generalized Poisson datasets generated by different solvers and at different matrix dimensions. The solver precision was set to 1e-12.
>
> | Gen. Method | Matrix Dim.         | Gen. Time         | NeurKItt Principal Angle Loss |
> | ----------- | ------------------- | ----------------- | ----------------------------- |
> | Eigsh       | 2500 / 6400 / 10000 | 10h / 80h / 800h  | 0.06 / 0.06 / 0.06            |
> | LOBPCG      | 2500 / 6400 / 10000 | 70h / 240h / 700h | 0.06 / 0.06 / 0.06            |
> | ChFSI       | 2500 / 6400 / 10000 | 16h / 44h / 160h  | 0.06 / 0.06 / 0.06            |
> | SCSF (ours) | 2500 / 6400 / 10000 | 9h / 26h / 45h    | 0.06 / 0.06 / 0.06            |
>
> *Note: To ensure a fair comparison, all data was interpolated to a 100x100* *grid* *before training.*
>
> The experiment shows that for matrix dimensions of 2500 and above, the choice of solver and the exact dimension have a negligible impact on the final trained model's performance. This is because the data precision is already far beyond what the neural network requires, confirming its status as a reliable "Ground Truth."
>
> **Action:** We will add the results of these supplementary experiments and a detailed analysis of discretization versus solver error to the final version of our paper.
>
> ### **On Performance Gain and Parallelization (W3)**
>
> - **Significance of the 3.5x Speedup:** The "3.5x" figure mentioned in the abstract is a conservative summary. As shown in **Table 1**, our performance gains are often much more substantial. For instance, on the Helmholtz dataset with 600 eigenvalues, SCSF achieves speedups of approximately **8x, 20x, 6x, and 95x** over Eigsh, LOBPCG, KS, and JD, respectively. This level of efficiency gain, which can shorten computation times from weeks to days, is of great practical importance for accelerating the scientific research cycle.
> - **Relationship with Parallelization:** SCSF and parallelization are **orthogonal and synergistic**. Our method accelerates the processing of a problem sequence on a single core. In practice, one can partition the N problems into M chunks and run SCSF in parallel on M cores, multiplying the speedup. All our experiments were conducted under fair, identical parallel conditions.
>
> **Action:** We will add a clear explanation to the final manuscript clarifying that our method is complementary to parallelization and that the reported speedups are achieved under fair, parallelized conditions.
>
> ###  **On Handling Boundary Conditions (Q1)**
>
> We use Dirichlet boundary conditions, incorporated during the matrix assembly stage, following standard procedures from benchmarks like [4-6].
>
> **Action:** We will clarify this in the revision and release our data generation code.
>
> ### **On the Effect of FDM Parameters on Error (Q2)**
>
> Our focus is on accelerating the matrix solve, not optimizing discretization parameters like grid spacing. To ensure fairness and relevance, we adopted discretization code from established benchmarks and tested our method across a wide range of matrix dimensions (2500 to 10000) to demonstrate SCSF's robustness and applicability in realistic dataset generation scenarios.
>
> ###  **On Other Comments:**
>
> Thank you for your constructive suggestions. We will revise Eq. (2) with a more illustrative 3x3 example, correct the typo on L86, move the bulk of Sec 2.2 to the appendix to improve flow, and expand our Limitations section to include a discussion on the intrinsic errors from discretization, as you recommended.
>
> **References:**
>
> [1] Spectral Inference Networks: Unifying Deep and Spectral Learning, ICLR 2019.
>
> [2] Neuralef: Deconstructing kernels by deep neural networks, NIPS 2022.
>
> [3] Operator SVD with neural networks via nested low-rank approximation, ICML 2024.
>
> [4] Fourier neural operator for parametric partial differential equations, ICLR 2019.
>
> [5] Neural krylov iteration for accelerating linear system solving, NIPS 2024.
>
> [6] Pdebench: An extensive benchmark for scientific machine learning, NIPS 2022.
>
> ---
> #### **Thanks again**
>
> We sincerely thank you again for your constructive and detailed feedback, which has helped us identify clear pathways to improve our paper. Should you have any further questions or require additional discussion, please don't hesitate to reach out. If we have adequately addressed your concerns, we would be grateful for your consideration in adjusting your evaluation score accordingly.

---

> > ### Comment · Reviewer_Hx22 · 2025-08-06
> >
> > The authors have made efforts to clarify my questions. The manuscript will be more readable and informative once these points are addressed in the revised version. Therefore, I encourage the authors to incorporate the related methods and discussions (including limitations we discussed above). I will update my rating to reflect it.

---

> ### Author Response · Authors · 2025-08-05
> **An error in our previous response**
>
> We sincerely apologize for the typographical error in our response to "**On the Significance of Accelerating Dataset Generation (W1)**"
>
> The last row of the table is misplaced. The correct version should be as follows
>
> | Category        | Algorithm          | Inference Time | Training Time | Dataset Gen. Time | Relative Residual |
> | --------------- | ------------------ | -------------- | ------------- | ----------------- | ----------------- |
> | Our Method      | SCSF (random init) | 1 min          | -             | -                 | 1e−8              |
> | Traditional     | Eigsh              | 1 min          | -             | -                 | 1e−8              |
> | Supervised NN   | NeurKItt           | 0.1s           | 3h            | 20h               | 1e−2              |
> | Unsupervised NN | NeuralEF           | 2h             | -             | -                 | 1e−2              |
> | Unsupervised NN | NeuralSVD          | 3h             | -             | -                 | 1e−2              |
>
> This was an oversight on our part during the format conversion process. We deeply regret any confusion this may have caused.

---

> > ### Comment · Reviewer_Hx22 · 2025-08-06
> >
> > I appreciate the new results presented in the table. However, I am not sure if I fully understand the results. In "Unsupervised NN", the eigenfunctions are also directly parameterized by neural nets, so why can the "Inference Time" be hours?

---

> > > ### Author Response · Authors · 2025-08-06
> > > **Response to Follow-up Questions from Reviewer Hx22 (1/2)**
> > >
> > > Dear Reviewer,
> > >
> > > Thank you for your follow-up question and for giving us the opportunity to clarify this important point. We sincerely appreciate your continued engagement and your willingness to help us improve the manuscript.
> > >
> > > ---
> > >
> > > ### **Clarification on "Inference Time" for Unsupervised** **Neural Networks**
> > >
> > > We understand your question regarding the "Inference Time" for unsupervised methods in our comparison table. We sincerely apologize for the ambiguity caused by our choice of terminology; we recognize that "Inference Time" was not precise in this context and we are grateful that you have pointed this out. Please allow us to provide a detailed clarification from the following perspectives.
> > >
> > > 1. **Fundamental** **Paradigm** **Difference in Solving Eigenvalue Problems**
> > >
> > > The core of the issue lies in the fundamentally different approaches that supervised and unsupervised neural networks take to solve eigenvalue problems:
> > >
> > >    - **Supervised Methods (e.g., NeurKItt [3]):** These methods are **data-driven**. They require a large, pre-generated dataset to train a model that learns a direct mapping from a given operator to its corresponding eigenpairs. Once this expensive, one-time training is complete, the "inference" stage is extremely fast: for a new, unseen operator, the model only needs to perform a single forward pass to predict the solution. This is why its inference time is very short (0.1s in our table).
> > > - **Unsupervised Methods (e.g., NeuralEF [1], NeuralSVD [2]):** These methods are **data-free**. They do not require any pre-existing labeled data. Instead, for each given operator, they solve for the eigenfunctions from scratch. This process is essentially a **non-linear** **optimization** **problem**: a neural network is iteratively trained by optimizing its parameters to minimize a specific objective function (e.g., a generalization of the Rayleigh quotient or a low-rank approximation error) until the network's output accurately converges to the true eigenfunction.
> > >
> > > Therefore, what we labeled as "Inference Time" for unsupervised methods is, more accurately, the **"Solving Time per Operator."** This multi-hour duration reflects the entire, iterative optimization and training process required to find the solution for a single operator. It is not a simple model inference but a complete, self-contained solution-finding procedure. This explains why the time is significantly longer than that of a trained supervised model, despite both using neural networks.
> > >
> > > 2. **Context from Prior Work and Algorithmic Efficiency**
> > >
> > > Furthermore, this performance characteristic is consistent with the established literature. For low-dimensional PDEs (e.g., 1 to 3 dimensions), traditional eigenvalue algorithms, which are based on linear iterative methods, converge much faster than unsupervised neural network methods, which rely on non-linear optimization. The primary advantage of unsupervised neural methods lies in their ability to tackle problems that are either too high-dimensional for traditional methods to handle due to memory constraints (e.g., 5 or 10 dimensions) or cannot be modeled by traditional algorithms. This point is also mentioned in related work on unsupervised eigenvalue solvers (e.g., Section 4.2.2, p. 15 of [4]).

---

> > > > ### Author Response · Authors · 2025-08-06
> > > > **Response to Follow-up Questions from Reviewer Hx22 (2/2)**
> > > >
> > > > 3. **Verification of Our Experimental Timings**
> > > >
> > > > To eliminate any remaining doubt, we have re-confirmed the 2-3 hour solving time we reported for NeuralEF [1] and NeuralSVD [2] and wish to provide the following details for transparency:
> > > >
> > > > - **Implementation:** All related experiments were conducted using the official code and training scripts provided in the NeuralSVD public GitHub repository. We only modified the operator definition and the number of eigenvalues to be solved to match our experimental setup. The reported solving time of 2-3 hours is consistent even for the original eigenvalue problems discussed in the NeuralSVD paper.
> > > > - **Hardware:** All experiments were timed on a single NVIDIA GeForce RTX 3090 GPU.
> > > > - **Comparison with Published Results:** While NeuralSVD does not report total solution times in their paper, NeuralEF [1] does. In Section 4.3 (p. 8, left column), the authors mention that solving for 10 eigenvalues of a matrix with a side length of $5×10^5$ takes half a day. In our experiments, we are solving for 100 eigenvalues of a matrix with a side length of $6.4×10^3$. Given these parameters, a solving time of 2-3 hours is entirely plausible and in line with expectations.
> > > >
> > > > We hope this detailed explanation fully addresses your question and clarifies the distinction between these methods. In the final version, we will add a discussion of unsupervised methods to our "Related Work" section.
> > > >
> > > > **References:**
> > > >
> > > > [1] NeuralEF: Deconstructing kernels by deep neural networks, NeurIPS 2022.
> > > >
> > > > [2] Operator SVD with neural networks via nested low-rank approximation, ICML 2024.
> > > >
> > > > [3] Neural Krylov iteration for accelerating linear system solving, NeurIPS 2024.
> > > >
> > > > [4] Neural networks based on power method and inverse power method for solving linear eigenvalue problems, Computers & Mathematics with Applications, 2023.
> > > >
> > > > ---
> > > >
> > > > We sincerely thank you once again for your detailed and constructive engagement with our work. Your meticulous review has been invaluable in helping us strengthen our paper. We are grateful for your time and guidance, and we would be deeply appreciative if you would consider our clarifications in your final evaluation. Should you have any further questions, please do not hesitate to let us know.

---

> > > > > ### Comment · Reviewer_Hx22 · 2025-08-07
> > > > >
> > > > > Thanks for the clarification. The table is useful, but I would suggest that the authors consider alternative names in illustrating the results. The "Inference Time" can be easily mistaken for the time to get the specific value of the eigenfunction at a given point, which should be ignorable in unsupervised NN-based methods. And in unsupervised NN, training itself is to learn the eigenfunction, which shall be included in "Solving Time per Operator."

---

> > > > > > ### Author Response · Authors · 2025-08-07
> > > > > > **Thanks for Your Feedback and Revision Plan**
> > > > > >
> > > > > > We greatly appreciate your constructive feedback and your willingness to raise your score!
> > > > > >
> > > > > > Thank you for the excellent suggestion regarding the terminology. We agree that "Inference Time" is ambiguous for unsupervised methods. As you recommended, we will revise the table and discussion in our final manuscript to use a more precise term like "Solving Time per Operator," clarifying that this encompasses the entire training/optimization process for a single problem instance.
> > > > > >
> > > > > > Thank you again for your thoughtful engagement and valuable guidance, which have significantly improved the clarity of our paper.

---

### Decision · Program_Chairs · 2025-09-17

**Decision:**

Reject

**Comment:**

Overall, while the reviewers definitely found certain merit in the paper, none were strictly positive about it, and thus aggregated consensus is that it is below bar for NeurIPS.